# Attenuation of phytofungal pathogenicity of *Ascomycota* by autophagy modulators

Jongchan Woo [1,2,3,11], Seungmee Jung[2,11], Seongbeom Kim [4], Yurong Li[5,9], Hyunjung Chung[4], Tatiana V. Roubtsova[6], Honghong Zhang[5,10], Celine Caseys [7], Dan Kliebenstein [7], Kyung-Nam Kim[8], Richard M. Bostock[6], Yong-Hwan Lee [4], Martin B. Dickman[5,12], Doil Choi [3] ✉, Eunsook Park [2] ✉ & Savithramma P. Dinesh-Kumar [1] ✉

Autophagy in eukaryotes functions to maintain homeostasis by degradation and recycling of long-lived and unwanted cellular materials. Autophagy plays important roles in pathogenicity of various fungal pathogens, suggesting that autophagy is a novel target for development of antifungal compounds. Here, we describe bioluminescence resonance energy transfer (BRET)-based high-throughput screening (HTS) strategy to identify compounds that inhibit fungal ATG4 cysteine protease-mediated cleavage of ATG8 that is critical for autophagosome formation. We identified ebselen (EB) and its analogs ebselen oxide (EO) and 2-(4-methylphenyl)−1,2-benzisothiazol-3(2H)-one (PT) as inhibitors of fungal pathogens *Botrytis cinerea* and *Magnaporthe oryzae* ATG4-mediated ATG8 processing. The EB and its analogs inhibit spore germination, hyphal development, and appressorium formation in *Ascomycota* pathogens, *B. cinerea*, *M. oryzae*, *Sclerotinia sclerotiorum* and *Monilinia fructicola*. Treatment with EB and its analogs significantly reduced fungal pathogenicity. Our findings provide molecular insights to develop the next generation of antifungal compounds by targeting autophagy in important fungal pathogens.

Annual crop losses are estimated to be USD 550 billion worldwide. About 40% of the total crop loss is due to plant diseases of which loss by fungal pathogens is about USD 150 billion[1]. The fungal pathogen, *Botrytis cinerea*, infects over two hundred plant species, causing loss of food resources over USD 10 billion annually. The rice blast fungus, *Magnaporthe oryzae*, destroys over 50 million tons of rice sufficient to feed one billion people annually[1]. Changes in fungal pathogen population dynamics due to climate change[2] and the emergence of fungicide-resistance of *B. cinerea* are major challenges in controlling fungal diseases[3]. Therefore, characterization of new molecular targets to control devastating fungal pathogens remains unmet needs for which effective and environmentally safe fungicides help ensure food security.

To develop new antifungal reagents against the devastating pathogens, it is of importance to select pivotal molecular targets

[1]Department of Plant Biology and the Genome Center, College of Biological Sciences, University of California, Davis, CA, USA. [2]Department of Molecular Biology, College of Agriculture, Life Sciences and Natural Resources, University of Wyoming, Laramie, WY, USA. [3]Plant Immunity Research Center, College of Agriculture and Life Sciences, Seoul National University, Seoul, Republic of Korea. [4]Department of Agricultural Biotechnology, College of Agriculture and Life Sciences, Seoul National University, Seoul, Republic of Korea. [5]Department of Plant Pathology and Microbiology, College of Agriculture and Life Sciences, Texas A & M University, College Station, TX, USA. [6]Department of Plant Pathology, College of Agriculture and Environmental Sciences, University of California, Davis, CA, USA. [7]Department of Plant Sciences, College of Agriculture and Environmental Sciences, University of California, Davis, CA, USA. [8]Department of Bioindustry and Bioresource Engineering, College of Life Sciences, Sejong University, Seoul, Republic of Korea. [9]Present address: Corteva Agriscience, Johnston, IA, USA. [10]Present address: Fujian University Key Laboratory for Plant-Microbe Interaction, College of Plant Protection, Fujian Agriculture and Forestry University, Fuzhou, China. [11]These authors contributed equally: Jongchan Woo, Seungmee Jung. [12]deceased: Martin B. Dickman. ✉e-mail: doil@snu.ac.kr; epark4@uwyo.edu; spdineshkumar@ucdavis.edu

involved in pathogenicity for successful screening of antifungal agents. Macroautophagy (hereafter referred to as autophagy) is a conserved process across eukaryotes that entails engulfment of cellular components or cargoes in double-membrane vesicles called autophagosomes. The autophagosomes are fused to the vacuoles/lysosomes for degradation and recycling[4–7]. Recycling of long-lived cellular proteins and organelles by autophagy is an important adaptive response to nutrient deprivation and fluctuation of environments. Autophagy participates in other diverse biological processes including cellular differentiation and development, cell and tissue homoeostasis, aging, senescence, innate and adaptive immunity, and programmed cell death (PCD)[4–6,8]. Over 30 autophagy-related genes have been identified in yeast and their homologs are well conserved across eukaryotes[6]. Among AuTophaGy (ATG) core components, ubiquitin-like ATG8 plays an important role in autophagosome initiation and biogenesis[9]. Furthermore, ATG8 serves as a docking site for adapter proteins and also plays a key role in autophagic cargo recruitment into autophagosomes[10]. The ATG4 cysteine proteases are important for maturation and recycling of ATG8s[4–6,8,11,12]. Ubiquitin-like ATG8s are cleaved at the catalytically important C-terminal glycine (Gly) residue by ATG4s. The Gly residue of the processed ATG8 is involved in adduct formation with phosphatidylethanolamine (PE)[13] after ubiquitin-like conjugation reaction carried out by E1-like ATG7, E2-like ATG3 and ATG10, and ATG5-12-16[14–17]. ATG8-PE on the outside membrane can be recycled for delivery of new lipid molecules after additional catalysis by ATG4s[4–6,8,12]. In contrast, the ATG8-PE in the inner membrane of the autophagosomes are degraded in the vacuolar/lysosomal lumen like autophagic cargoes[18–20]. ATG8-PE integrates into the phagophore, and it marks the autophagosome until vacuolar fusion and cargo delivery. Because of the association of ATG8 during autophagosome maturation, a fluorescence protein tagged with an ATG8 such as GFP-ATG8 is presently the most widely used marker for visualizing autophagosomes[21,22].

*ATG* genes are also conserved in the sequenced plant pathogenic fungi. Recent studies indicate that autophagy plays important roles in homeostasis, cellular differentiation, nutrient starvation, sporulation, and pathogenicity of fungal pathogens[23–31]. During host-fungal interactions, conidia germination of pathogenic fungi is an initial step of the infection cycle followed by germ tube formation to find an infection site on the host surface for successful invasion. The deletion of genes encoding core components of autophagy in *B. cinerea* showed loss of pathogenicity and defect in fungal life cycle[25,29,30]. For example, the genetic defects of *BcATG8* and *BcATG4* blocked autophagy, resulting in significant impairment of vegetative development and pathogenicity of the fungal pathogen[25,29]. Similar phenotypes of autophagy mutants have been observed in another devastating fungal pathogen, *M. oryzae*. Autophagy in *Moatg4* and *Moatg8* mutants was severely impaired and the virulence was attenuated[23,26], indicating that autophagy could play a pivotal role in fungal pathogenicity. Once successful engagement between the fungus and its host, development of an infectious structure such as appressorium is initiated and followed by autophagic cell death[32]. Intriguingly, the phenotypes of *Moatg4* and *Moatg8* mutants indicate that *Mo*ATG4 and *Mo*ATG8 contribute not to formation of the structures but to proper function and/or maturation of appressoria[23,26]. Malfunctional appressoria of the autophagy mutants of *M. oryzae* cause to attenuate pathogenicity of the rice blast fungus[23,26]. The importance of autophagy for fungal pathogenicity from the above discussed studies indicates that autophagy is an excellent target to control fungal pathogens.

Here, we describe a novel strategy to identify chemicals that interfere with fungal autophagy and hence attenuate its pathogenicity. To identify chemical modulators of fungal autophagy, we generated a BRET-based sensor of *Bc*ATG8 (*Bc*ATG8-sensor) and optimized the target-based high-throughput screening (HTS) for the *Bc*ATG4-mediated processing of the *Bc*ATG8-sensor. We identified ebselen (EB), ebselen oxide (EO), and 2-(4-methylphenyl)−1,2-benzisothiazol-

3(2H)-one (PT) as inhibitors of ATG8 processing by ATG4 in *B. cinerea* and *M. oryzae*. Our results show that these chemicals significantly inhibit pathogenicity of *B. cinerea* and *M. oryzae* as well as other important pathogens from *Ascomycota*. Taken together, the target-based HTS with the BRET-based synthetic ATG8-sensors to identify modulators of autophagy is an excellent strategy to identify novel chemicals for controlling devastating phytofungal pathogens.

## Results
### Target-based HTS using the BRET-based ATG8-sensor to identify fungal autophagy inhibitors
Autophagosome biogenesis requires the ATG4 protease-mediated cleavage of ATG8 at the conserved C-terminal Gly residue[4–6,8,12]. We previously reported that a BRET-based synthetic substrate of *Arabidopsis* ATG8a mimics endogenous *At*ATG8a[33,34]. This BRET-based ATG8-sensor, in combination with in vitro cleavage assay, prompted us to optimize target-based HTS[35] for the identification of new autophagy modulators against agronomically important fungal pathogens (Fig. 1a). *B. cinerea* and *M. oryzae* are among the top ten devastating fungal pathogens[36]. Since autophagy plays an important role in pathogenicity of these devastating fungal pathogens[23,25,26,29,30], we designed and generated the BRET-based *B. cinerea* ATG8-sensor by fusing the citrine fluorescent protein at the N-terminus and the modified *Renilla* luciferase superhLUC (ShR)[37] at the C-terminus of *Bc*ATG8 (Citrine-*Bc*ATG8-ShR; hereafter referred to as *Bc*ATG8-sensor) (Fig. 1a). We used the *Bc*ATG8-sensor to optimize HTS for the *Bc*ATG4-mediated processing of *Bc*ATG8 (Fig. 1a and Supplementary Fig. 1a). In this assay, cleavage of the *Bc*ATG8-sensor by *Bc*ATG4 leads to separation of Citrine-*Bc*ATG8 and ShR and hence results in a low BRET ratio. In contrast, if *Bc*ATG4 fails to cleave the *Bc*ATG8-sensor, it will result in a high BRET ratio. Consistent with this, the BRET ratio was high when the non-cleavable *Bc*ATG8 (G116A) mutant-sensor was used in the assay compared to the wild type *Bc*ATG8-sensor (Supplementary Fig. 1a). These results indicate that hit compounds to control the *Bc*ATG4-mediated processing of *Bc*ATG8 can be easily identified by monitoring BRET ratios from the in vitro cleavage assay with the *Bc*ATG8-sensor and the recombinant *Bc*ATG4 in the presence of the chemical modulators. In addition, we examined the sensitivity of our HTS platform. We observed a statistically significant difference of BRET ratio between the uncleaved *Bc*ATG8-sensor and the *Bc*ATG8-sensor processed by *Bc*ATG4 as low as -1.6 ng (177.3 pM) (Supplementary Fig. 1b). The detection capability of our HTS with the pico-molar range suggested that the *Bc*ATG8-sensor could be used in HTS to identify chemicals that might inhibit or enhance the cleavage by *Bc*ATG4. Using HTS, we screened 2,485 chemicals that included biologically active and clinically evaluated small molecules. Two compounds, 9F4 (Pubchem CID, 675983) and an organoselenium compound ebselen (EB; Pubchem CID, 3194), were identified as the inhibitors and their inhibitory roles were further validated by in vitro cleavage assay (Supplementary Fig. 1c, d). The detection sensitivity for EB was -0.3 μM which was sufficient to inhibit the cleavage of -3 ng of the *Bc*ATG8-sensor (337 pM) by -6 ng of *Bc*ATG4 (1.12 nM) in our HTS (Supplementary Fig. 1e). Together, our target-based screening platform with the BRET-based *Bc*ATG8-sensor is highly sensitive, compared to other cell-based HTS[38,39]. In our HTS, 30 compounds inhibited the *Bc*ATG4-mediated processing of *Bc*ATG8, while 14 compounds activated the processing (Supplementary Fig. 2). Due to strong inhibition of EB compared to 9F4 (Fig. 1b and Supplementary Fig. 1c, d), EB was chosen for further study.

Since selenium in EB is known to react with a thiol group in a reversible manner[40], we hypothesized that the decoration on thiol groups in *Bc*ATG4 and/or *Bc*ATG8 bound by selenium in EB could inhibit the *Bc*ATG8 cleavage. To demonstrate if selenium in EB could serve as the functional group of EB, we selected analog compounds such as ebselen oxide (EO), 2-(4-methylphenyl)−1,2-benzisothiazol-3(2H)-one (PT), 2-phenyl-3H-indol-3-one (PIO), and 2-phenyl-1H-

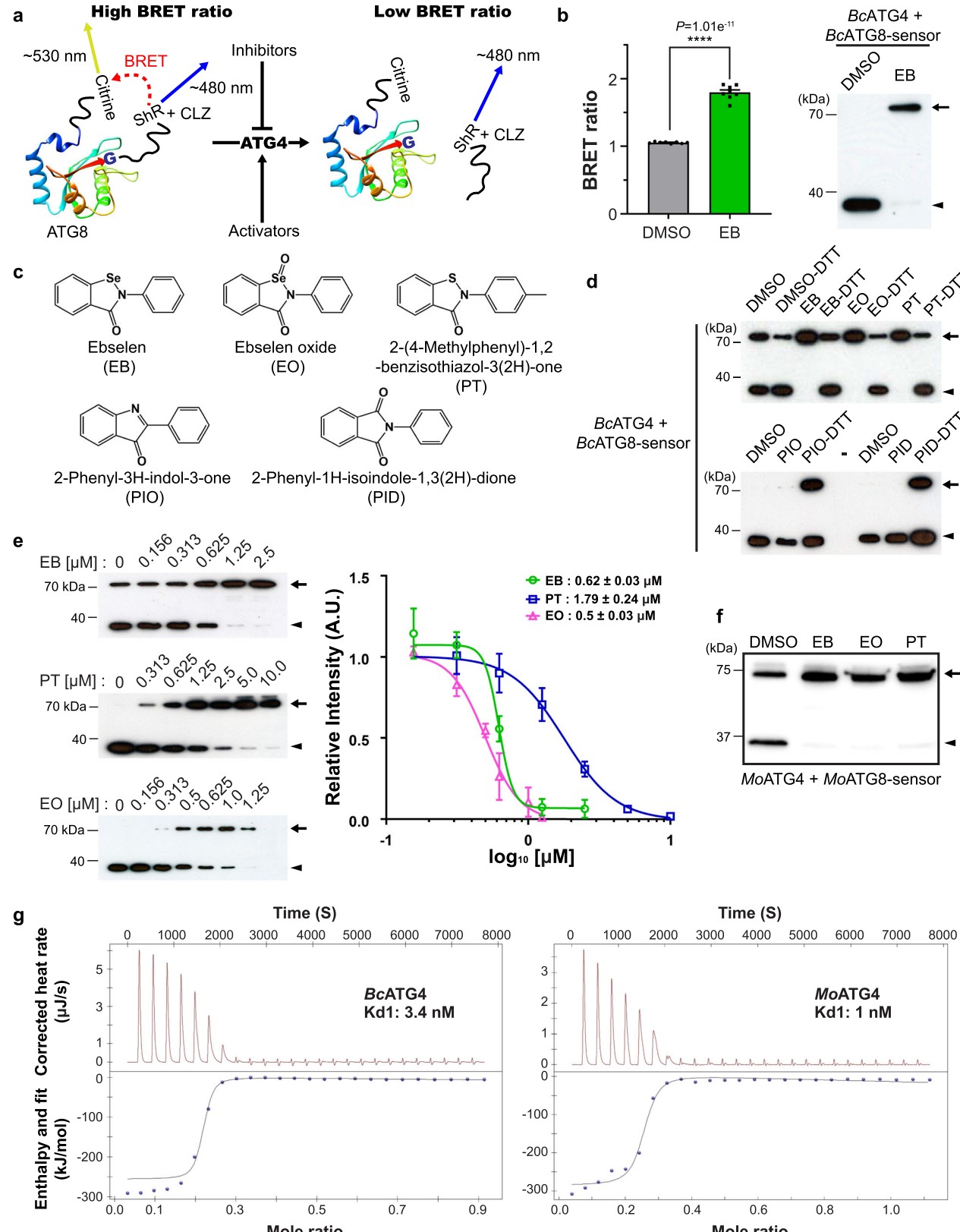

isoindole-1,3(2H)-dione (PID) based on the chemical structure of EB (Fig. 1c). Compared to PIO and PID, EO and PT analogs strongly inhibited the cleavage of BcATG8 by BcATG4 similar to EB (Fig. 1d). The loss of the inhibitory effect of PID is underpinned by the non-reactive carbonyl group in the functional group because the chemical scaffold of PID is the same as that of EO except the functional group (Fig. 1c, d).

The inhibition of the BcATG8 cleavage by EB, EO, and PT was suppressed in the presence of DTT, indicating that DTT could contribute to reduction of the selenium-sulfur[40] and the disulfide bond[41] between the inhibitors (EB, EO, and PT) and the polypeptides in the cleavage process (Fig. 1d). Therefore, the selenium in EB, the oxidized selenium in EO, and the sulfur in PT reacted with a thiol can serve as the

**Fig. 1 | Characterization of autophagy modulators that target fungal ATG4-mediated cleavage of ATG8. a** Schematics of HTS using the bioluminescence resonance energy transfer (BRET)-based *B. cinerea* synthetic ATG8 (*Bc*ATG8)-sensor in which citrine fluorescence protein and modified *Renilla* luciferase, Super-hRLUC (ShR), are fused to the N- and the C-terminus of *Bc*ATG8, respectively. *Bc*ATG4 cysteine protease-mediated cleavage of Citrine-*Bc*ATG8-ShR leads to generation of Citrine-*Bc*ATG8 and ShR byproducts. Citrine-*Bc*ATG8-ShR exhibits a higher BRET ratio in the presence of ShR substrate coelenterazine (CLZ) and the cleavage of the sensor leads to lower BRET ratio. Compounds with high and low BRET ratios are considered as inhibitors and activators, respectively. **b** BRET ratio of the ebselen (EB) treatment (left). The cleavage of Citrine-*Bc*ATG8-ShR by *Bc*ATG4 (right) confirms the inhibition of the *Bc*ATG4-mediated *Bc*ATG8 processing by EB compared to the DMSO control. The graph presents mean and SE. $p < 0.0001$ (****),

two-tailed *t*-test. **c** Chemical scaffolds of EB and its analogs, EO, PT, PID, and PIO. **d** EO and PT inhibit the *Bc*ATG8 cleavage similar to EB. PID and PIO analogs of EB have weak effects on the cleavage. Inhibition of the cleavage is reversible under reducing conditions (+DTT). **e** $IC_{50}$ values of EB, EO, and PT for the *Bc*ATG8 cleavage are estimated by in vitro cleavage assay (left panels). The graph (right) shows mean and SE. **f** The inhibition of the *M. oryzae* ATG4 (*Mo*ATG4)-mediated processing of *Mo*ATG8 by EB, EO, and PT. **g** The equilibrium dissociation constant of EB is estimated by ITC. The ITC data of EB with *Bc*ATG4 (left) and *Mo*ATG4 (right) are fitted to the model of multiple binding sites. Three independent experiments were performed with similar results in **b**–**g**. A representative image is shown in each panel. Arrows and arrowheads represent the full-length BRET-sensors and the cleavage byproducts, respectively.

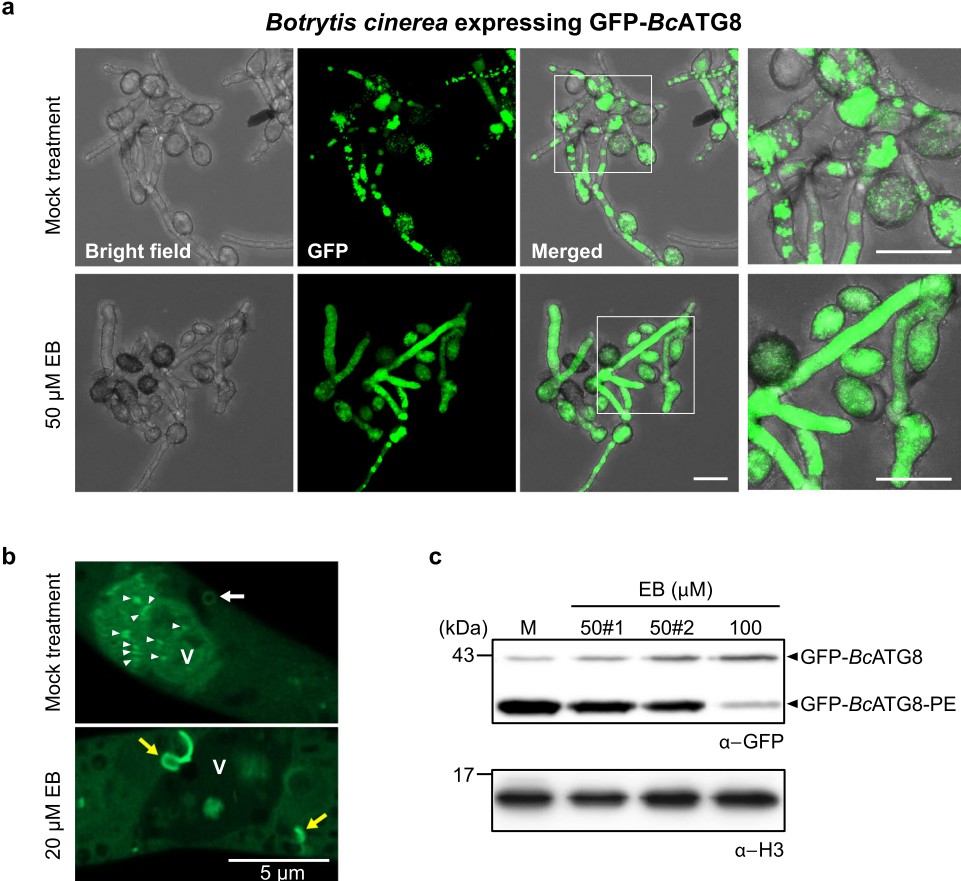

**a**

***Botrytis cinerea* expressing GFP-*Bc*ATG8**

**b**

**c**

**Fig. 2 | EB inhibits autophagy in *B. cinerea*. a** In vivo inhibition of autophagy by EB using transgenic *B. cinerea* expressing GFP-*Bc*ATG8. GFP localization pattern in *B. cinerea* expressing GFP-*Bc*ATG8 in the mock control or the EB treatment under nutrient starvation, an autophagy inducing environment. GFP puncta indicative of autophagosomes are observed in mock treatment (top second panel), compared to diffusion of GFP fluorescence in the cytoplasm under EB treatment (bottom second panel). Right panels are magnifications of white boxed areas of each panel. Scale bars, 20 μm. **b** Super-resolution images of incomplete autophagosomes under EB treatment. In mock treatment, the complete autophagosomes (white arrow) are observed while the EB-treated fungal cells show incomplete membranous structures stained with GFP-*Bc*ATG8 (yellow arrows) frequently under 20 μM EB

treatment. White arrowheads indicate autolysosomes in the vacuole (V). Scale bar, 5 μm. **c** Autophagy inhibition by detecting level of lipidation on GFP-*Bc*ATG8 under EB treatment. The fully grown transgenic *B. cinerea* expressing GFP-*Bc*ATG8 is treated with either 50 μM or 100 μM EB for 8 hrs and proteins are separated in 6 M urea SDS-PAGE. The lipidation on GFP-*Bc*ATG8 is reduced whilst the unmodified GFP-*Bc*ATG8 accumulates under EB treatment, compared to the mock (M) control (top panel), indicating that EB can inhibit autophagy. #1 and #2 of 50 μM EB represent technical replicates. The blot probed with α-H3 is used as a loading control (bottom panel). The three biologically independent experiments were conducted with similar results in **a**–**c**. A representative image is shown in each panel.

functional group because the nitrogen in PIO and the carbonyl group in PID have weak or no binding with a thiol group. Based on the $IC_{50}$ values (Fig. 1e), EB and EO are stronger inhibitors than PT for inhibition of the *Bc*ATG4-mediated cleavage of *Bc*ATG8 in vitro. The different inhibitory activities of the lead compounds appear to result from the different functional groups in the chemical scaffolds. In addition,

higher $IC_{50}$ of PT suggests that the sulfur in the benzisothiazol moiety and the methyl modification in the 2-4-methyphenyl moiety may attenuate the efficacy of the inhibitor in vitro (Fig. 1e). The antioxidant function of EB[42,43] appears not to be directly involved in the inhibition of the *Bc*ATG8 cleavage in vitro because EO has no antioxidant activity[43].

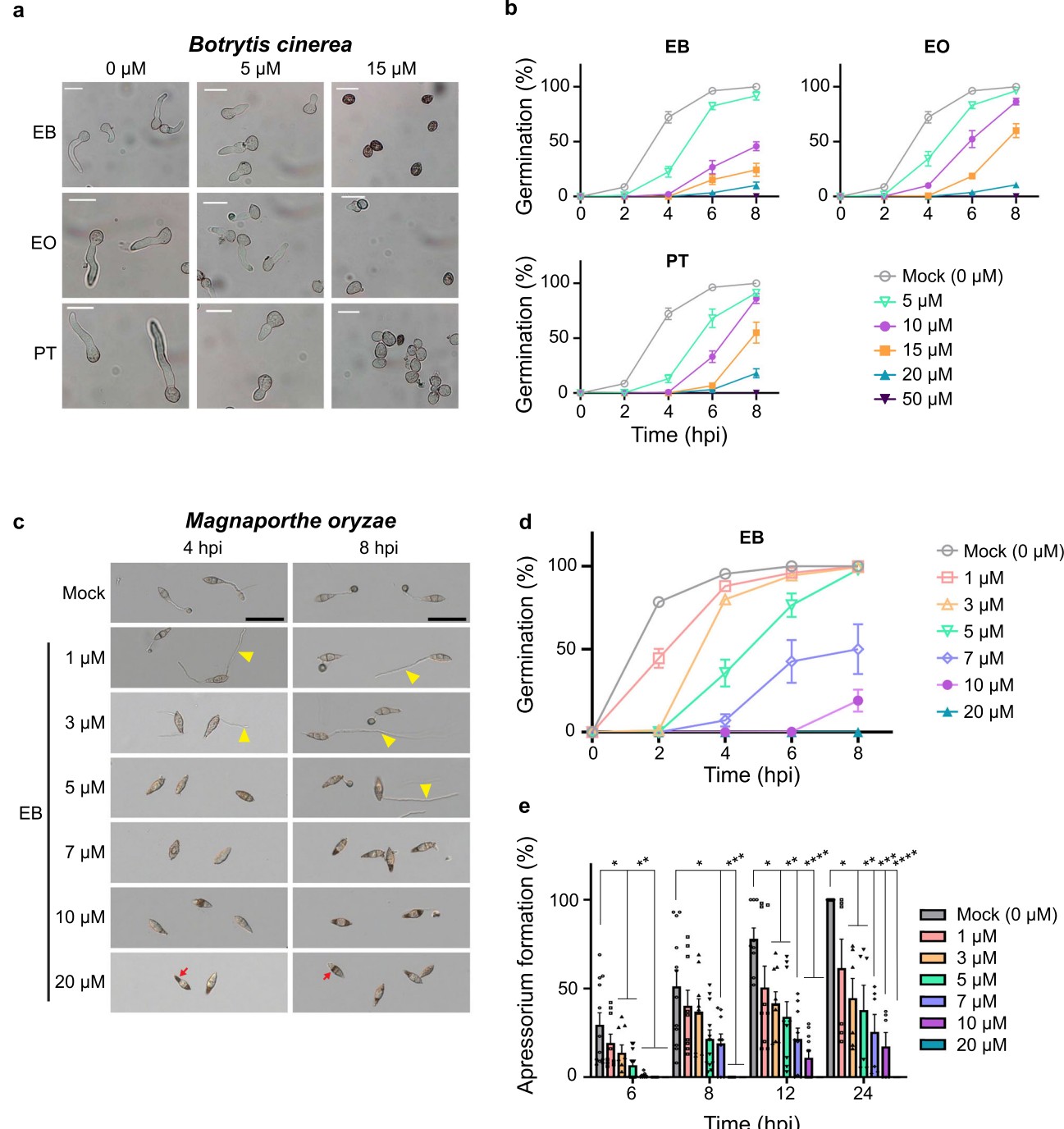

**Fig. 3 | EB, EO, and PT inhibit the conidial germination of _B. cinerea_ and _M. oryzae_ in a dose-dependent manner. a** Morphological differences of germinating conidia of _B. cinerea_ in the presence of different concentrations of autophagy inhibitors EB, EO and PT at 6 hpi. Dead conidia are often observed at 15 μM and no germination is observed at 50 μM of all compounds. _n_ = 3 biologically independent experiments. A representative image is shown. Scale bars, 20 μm. **b** Germination percentages of _B. cinerea_ in the presence of indicated concentrations of autophagy inhibitors EB, EO and PT, compared to the mock control at different time points. All chemicals tested significantly reduce germination at concentrations above 15 μM. The statistical analyses are summarized in Supplementary Table 1. _n_ = 3 biologically independent experiments. **c** Suppression of conidial germination and appressorium formation of _M. oryzae_ under EB treatment. Photographs were taken at 4 and

8 hpi. Yellow arrowheads and red arrows indicate abnormal hyphae and dead conidia, respectively. The experiments were repeated three times and similar results were observed. A representative image is shown. Scale bars, 50 μm. **d** The delay in germination of _M. oryzae_ is observed at low concentrations of EB and the germination rate is significantly reduced above 7 μM EB. The statistical analyses are summarized in Supplementary Table 2. _n_ = 3 biologically independent experiments. **e** Significant suppression of appressorium formation of _M. oryzae_ is observed above 10 μM EB. All graphs display means with SE. Two-way ANOVA test with Dunnett's multiple comparison. A different number of asterisks indicates statistically significant differences. _n_ = 3 biologically independent experiments. Summary of the statistical analyses is in Supplementary Table 3.

Since autophagy plays essential roles in the life cycle of *M. oryzae*[23,26,27,32], we also tested if EB and its analogs could suppress the *Mo*ATG4-mediated processing of *Mo*ATG8. We observed strong inhibition of the *Mo*ATG4-mediated cleavage of *Mo*ATG8 by EB and its analogs (Fig. 1f). Isothermal Titration Calorimetry (ITC) experiments validated that EB could directly bind to *Bc*ATG4 and *Mo*ATG4 with the degree of dissociation strength ($K_d$) of 3.4 and 1 nM, respectively (Fig. 1g). The low $K_d$ of EB indicates the higher affinity to ATG4s than EO and PT (Supplementary Fig. 3a–d). However, we were unable to obtain titration data with *Bc*ATG8, suggesting that EB and its analogs could not bind to *Bc*ATG8. Together, these results indicate that EB and its analogs EO and PT inhibit *Bc*ATG4- and *Mo*ATG4-mediated cleavage of *Bc*ATG8 and *Mo*ATG8, respectively.

## EB inhibits autophagy in *B. cinerea*

To monitor EB inhibitory effect on the *Bc*ATG8 cleavage in vivo, we generated *B. cinerea* expressing GFP-*Bc*ATG8. Under autophagy inducing conditions in mock treatment, we observed many GFP puncta representing autophagosomes (Fig. 2a, top panels). However, EB treatment (50 μM) even under the autophagy inducing condition resulted in a diffuse GFP fluorescence signal in the fungal cells (Fig. 2a, bottom panels). Since unprocessed GFP-*Bc*ATG8 cannot be incorporated into autophagosomes[4–6,11], the observed diffuse pattern of GFP-*Bc*ATG8 in the cytoplasm upon EB treatment indicates that GFP-*Bc*ATG8 is not processed. The inhibition of autophagy by EB was further confirmed by a super-resolution microscopy. Since the ATG4-mediated cleavage of ATG8 is important for delivery of a membrane lipid such as PE[13–17], the inhibition of the enzymatic process causes not only incompletion of autophagosome formation in the cytoplasm but also less accumulation of autolysosomes in the vacuole. In the high magnification microscopic images, we observed both abnormal membrane structures stained with GFP-*Bc*ATG8 in the cytoplasm and less autolysosomes in the vacuole under EB treatment, compared to mock treatment (Fig. 2b). These results indicate that the *Bc*ATG4 inhibition by EB causes incomplete autophagy, resulting in less accumulation of autolysosomes in the fungal cells. We also examined transcript levels of known autophagy core components (*Bc*ATG1, *3*, *4*, *6*, *7*, and *8*) in *B. cinerea* under the same condition which caused abnormal formation of autophagosomes observed under the microscopy. None of the tested *Bc*ATG transcript levels changed upon EB treatment, indicating that EB had no effect on the transcription of known *B. cinerea* autophagy core components (Supplementary Fig. 4a).

Once ATG8 is processed by ATG4, the exposed Gly residue in the C-terminus of ATG8 is involved in adduct formation with a membrane lipid, PE[13]. The lipidation status on ATG8 is a hallmark for autophagy biogenesis[17]. In addition, the lipidation of GFP-ATG8 has also been reported and widely used as an autophagy marker in various eukaryotes[44–46]. To further validate that EB inhibits autophagy in the fungal cells (Fig. 2a, b), we performed a pulse-chase experiment to monitor the lipidation of GFP-*Bc*ATG8 under EB treatment. Since the lipidated GFP-ATG8 migrates faster than the unmodified GFP-ATG8 in 6 M urea SDS-PAGE[44–46], we analyzed the lipidation status of GFP-*Bc*ATG8 under EB treatment for 8 h. The post-translational modification on GFP-*Bc*ATG8 was significantly inhibited in the presence of 50 μM EB and the inhibition was more severe in 100 μM EB, compared to mock treatment (Fig. 2c, top panel, bottom band). In parallel, we observed increased accumulation of the unmodified GFP-*Bc*ATG8 under EB treatment (Fig. 2c, top band) indicating the suppression of the GFP-*Bc*ATG8 maturation. Together, these results indicate that EB inhibits the *Bc*ATG4-mediated processing of GFP-*Bc*ATG8 in vivo.

Since EB inhibited autophagy that plays an important role in the fungal pathogenicity, we examined the effect of the inhibitors on growth of the transgenic *B. cinerea* expressing GFP-*Bc*ATG8 on tomato leaves. EB and its analogs significantly limited the growth of *B. cinerea*

expressing GFP-*Bc*ATG8, indicating that the autophagy inhibition by the lead compounds could compromise pathogenicity of the transgenic *B. cinerea* (Supplementary Fig. 4b). Since genetic variation in *B. cinerea* isolates often contribute towards host incompatibility[47,48], we tested if various isolates respond differently to EB. Severe impairment of conidial germination was observed under 20 μM and exacerbated under 50 μM EB in nine isolates of *B. cinerea* (Supplementary Fig. 5). Thus, targeting autophagy is an excellent strategy to attenuate the fungal growth regardless of the genetic variation of *B. cinerea*.

## EB, EO, and PT suppress the growth of *Ascomycota* fungal pathogens

We tested if EB, EO, and PT had broad effects on development and pathogenicity of *Ascomycota* pathogens including *B. cinerea*, *M. oryzae*, *Sclerotinia sclerotiorum*, and *Monilinia fructicola* since autophagy plays important roles in growth, hyphal development, and pathogenicity of *Ascomycota* pathogens[23,25–27,29–32]. Interestingly, *M. fructicola* failed to germinate in the presence of 20 μM EB and EO while the germination of *B. cinerea* and *M. oryzae* were abolished under 50 μM EB and EO (Supplementary Fig. 6a). It has been shown that potato PKI1 and PPI3B2 protease inhibitors inhibit conidia germination of *B. cinerea* but direct targets of these inhibitors are unknown[49]. In addition, the genetic defect of *Bc*ATG4 causes reduced germination[25]. These findings indicate that the function of proteases is important during germination of the fungal pathogen. Thus, we further tested germination rates of *B. cinerea* under the lead compounds in a time course. The conidia germinated after 6 h in water. However, conidial germination was delayed in 5 μM EB, EO, and PT and abolished at higher concentration of the inhibitors (Fig. 3a, b). Therefore, EB and its analogs could efficiently inhibit protease activity, resulting in suppression of germination of *B. cinerea*. We applied these experimental conditions to investigate if infection of *B. cinerea* could be inhibited on leaves of *Nicotiana benthamiana*. Based on the result from the germination test of *B. cinerea* under EB treatment (Fig. 3b), the conidia suspension of *B. cinerea* was directly mixed with low concentrations of EB (either 5 or 10 μM) and then inoculated on *N. benthamiana*. After 3 days post-inoculation (dpi), the disease symptom in *N. benthamiana* was monitored. Infection of *B. cinerea* was suppressed under 10 μM EB treatment, indicating a low-dose of EB could be effective to inhibit infection of *B. cinerea* on live plant tissues (Supplementary Fig. 6b).

With *S. sclerotiorum*, only 22% of the ascospores germinated under 50 μM EB at 12 h post-incubation (hpi) (Supplementary Fig. 6c). It has been reported that *S. sclerotiorum* ATG8 (*Ss*ATG8) directly interacts with *Ss*NBR1, a known autophagy receptor. Furthermore, the vegetative growth of the *Ssatg8* mutant was inhibited in potato dextrose agar (PDA)[50]. Thus, *Ss*ATG8 is a bona fide core component in autophagy biogenesis of the *S. sclerotiorum* pathogen. Based on these reports, we postulated that the inhibition of *Ss*ATG8 maturation could suppress autophagy, resulting in the growth retardation of the fungal pathogen under EB treatment. In particular, *Ssatg8* might be more resistant to EB than wild type in the growth inhibition because EB could inhibit ATG8 maturation in fungal pathogens. We evaluated the hyphae growth of *Ssatg8* at two time points of 40 hpi and 65 hpi because *Ssatg8* hyphae grew slowly, compared to wild type. *Ssatg8* was resistant to EB treatment at both time points whereas the significant growth inhibition by EB was observed in wild type (Supplementary Fig. 6d), indicating that the growth retardation of wild type *S. sclerotiorum* could be attributed to the inhibition of the *Ss*ATG8 processing by EB treatment. Therefore, the suppression of autophagy by EB resulted in the growth inhibition of wild type *S. sclerotiorum*.

A conidium of *M. oryzae* consists of three cells and typically produces a single germ tube from the apical cell during germination[32,51]. At the tip of a germ tube, a dome-shaped melanin-pigmented appressorium forms and penetrates rice cells[32,51]. Autophagy plays a pivotal role in the appressorial maturation of *M. oryzae*[23,26,32]. Therefore, we

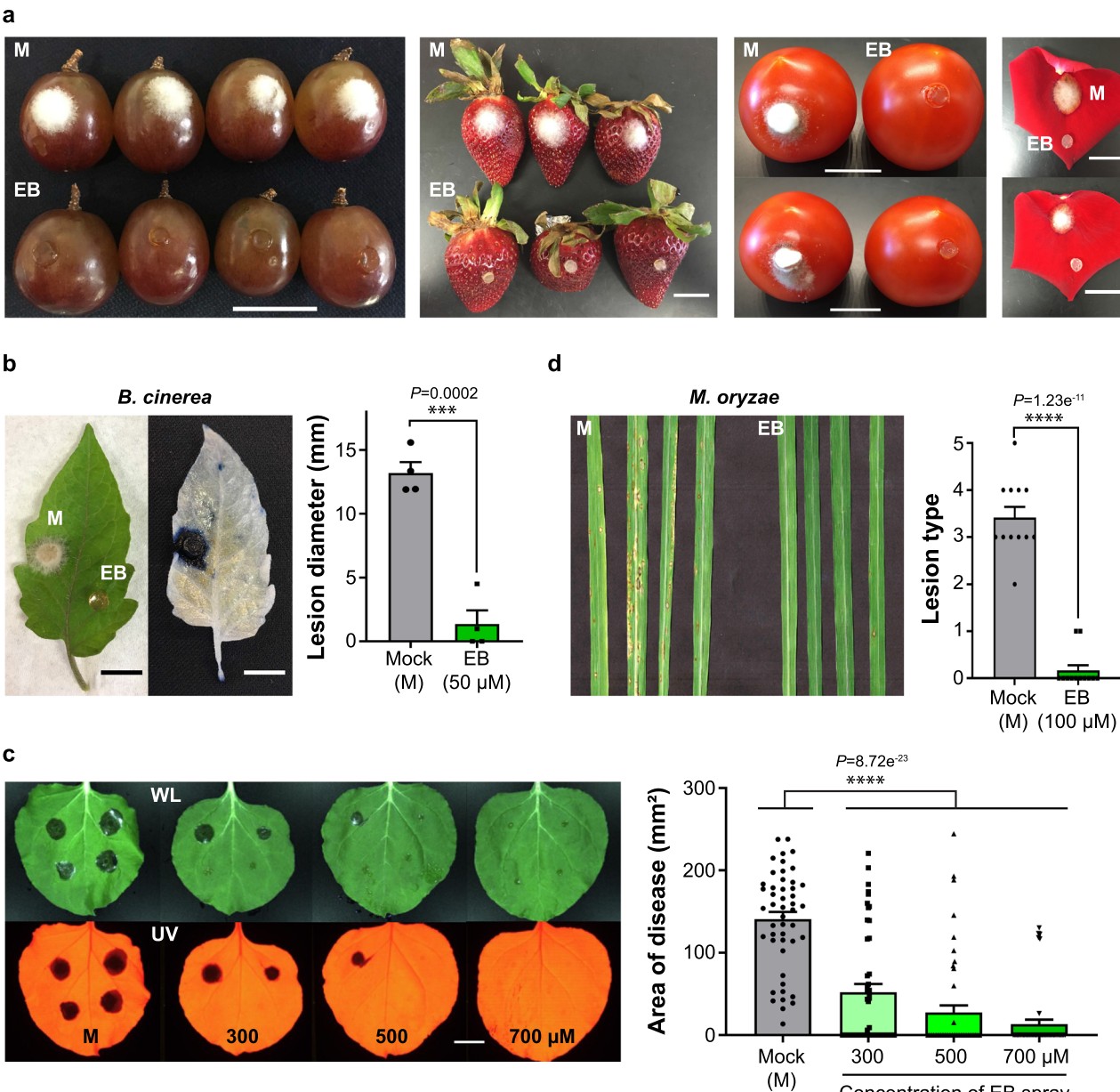

**Fig. 4 | EB inhibits infection of fungal pathogens in *Ascomycota* on various host plants. a** EB (50 μM) inhibition of *B. cinerea* lesion development on grapes, strawberries, tomatoes, and rose petals, compared to the mock (M) control. Three repeat experiments were performed with similar results. A representative image is shown. **b** Inhibition of *B. cinerea* infection on tomato leaves by EB, compared to the mock (M) control. Trypan blue staining of leaves shows the region of necrosis. The corresponding image of the live leaf is shown on the left. Mean lesion diameters in mock- and EB-treated leaves are plotted (right panel). Error bar, SE. $p = 0.0002$ (***), two-tailed *t*-test with Welch's correction. $n = 4$ biologically independent experiments. **c** Spray application of EB inhibits the growth of *B. cinerea* on *N. benthamiana* leaves. Each leaf image was taken under white light (WL) and UV light after 3 dpi. Lesion areas are plotted (right panel). Error bar, SE. $p < 0.0001$ (****), Dunnett's multiple comparisons test with one-way ANOVA. $n = 3$ biologically independent experiments. **d** Inhibition of *M. oryzae* by EB on rice plants. The conidial suspension of *M. oryzae* was mixed with either mock (M, 1% DMSO) or 100 μM of EB and directly applied onto rice plants. Lesion types were defined after a week. Error bar, SE. $p < 0.0001$ (****), two-tailed *t*-test. All scale bars in images, 2 cm. $n = 3$ biologically independent experiments.

examined the effects of the autophagy inhibitors EB, EO, and PT on early development of *M. oryzae*. A significant delay and reduction in germination was observed under EB treatment in a dose-dependent manner (Fig. 3c, d) as shown as the decreased germination rate of the mutant of *Moatg4*[26]. *M. oryzae* was more sensitive to EB than *B. cinerea* because 20 μM EB completely inhibited the germination of *M. oryzae* at 8 hpi (Fig. 3d). Within 4 hpi, one or two cells of three cells consisting of a conidium were darker in 20 μM EB treatment (Fig. 3c, red arrows). Propidium iodide staining revealed that these cells lost their viability (Supplementary Fig. 7). In the control, almost all conidia germinated and initiated appressorium formation by 24 hpi in our experimental

condition (Fig. 3d, e). Under 1–5 μM EB, *M. oryzae* developed abnormal conidia with long and/or bi-directional germ tubes (Fig. 3c, yellow arrowheads). Interestingly, appressorium formation was also inhibited in a dose-dependent manner (Fig. 3e). EO at low concentration also induced the morphological changes observed in EB treatment (Supplementary Fig. 8a, yellow arrowheads). However, EO showed different efficacies for the inhibition of germination and appressorium formation of *M. oryzae*, compared to EB treatment (Supplementary Fig. 8b, c). The phenotypes induced by PT were as similar to those observed by EO in *M. oryzae* (Supplementary Fig. 8d–f). In addition to the inhibition of both germination and appressorium formation of *M. oryzae*,

mycelial growth of *B. cinerea*, *M. fruticola*, and *S. sclerotiorum* was significantly reduced under 50 μM EB (Supplementary Fig. 9a). Similar to impaired sclerotial development of *atg8* mutants of both *B. cinerea* and *S. sclerotiorum*[29,50], 20 μM EB and EO significantly inhibited sclerotial development of *S. sclerotiorum* while 20 μM PT affected the development of sclerotia less than the others (Supplementary Fig. 9b). There appears to be different stability and permeability of PT due to the long-term incubation and different cell wall composition of sclerotia with that of mycelia. Together, these results show that treatment with new autophagy inhibitors EB, EO, and PT suppresses germination, mycelial development, appressorium formation, and sclerotial development in the fungal pathogens examined.

### Autophagic activity in fungal pathogens is essential for pathogenicity on the host plants

We assessed the inhibitory activities of EB and its analogs on the fungal infections in several hosts since the fungal mutants of *atg4* and *atg8* showed significant reduction of virulence[23,25,26,29,50]. Considering the agronomical and the holticultural importance, we selected host plants such as grapes, strawberries, tomatoes, and rose petals for fungal infection. EB treatment abolished *B. cinerea* lesion development on the hosts, compared to the mock-treated samples (Fig. 4a). Furthermore, EB treatment significantly inhibited *B. cinerea* lesion development on a vegetative tissue such as tomato leaves (Fig. 4b). In the case of *M. fructicola* where 20 μM EB completely suppressed the spore germination (Supplementary Fig. 6a), 20 μM EB also efficiently inhibited the growth of *M. fructicola* on apple fruits (Supplementary Fig. 10a). With 50 μM EB treatment, we observed significant reduction of *S. sclerotiorum* infection on tomato leaves (Supplementary Fig. 10b). To mimic typical fungicide application setting for investigation of the protective activity of EB against *B. cinerea*, we sprayed EB onto *N. benthamiana*, allowed it to dry, and then inoculated the conidia suspension of *B. cinerea* (Fig. 4c). The inhibition of lesion development on *N. benthamiana* leaves was monitored after spray application of EB following recent studies testing active ingredients of commercial fungicides such as difenoconazole and hexanoic acid against *B. cinerea*[52,53]. Dose-dependent inhibition of lesion development of *B. cinerea* was observed on *N. benthamiana* leaves infected by the conidia suspension (Fig. 4c). In addition, we examined the curative activity of EB in the live tissues. Prior to spray-application of EB, the conidia suspension of *B. cinerea* was inoculated and the disease symptom was incurred on the host tissues at 36 hpi. After development of the disease symptom, 300 μM EB was applied directly onto infected leaves once a day. The disease lesions were measured at 3 and 4 dpi. Compared to mock treatment, spray-application of EB showed significant inhibition of the disease progression, suggesting EB might have the curative activity against the necrotrophic fungal pathogen (Supplementary Fig. 10c). For *M. oryzae*, application of 100 μM EB was sufficient to abolish the rice blast infection by the conidia suspension while severe blast disease was developed in the mock-treated rice plants (Fig. 4d). Together, our results indicate that autophagy inhibition by EB, EO, and PT compounds can be effective in reducing the pathogenicity of *M. oryzae*, *B. cinerea*, *S. sclerotiorum*, and *M. fructicola* on respective hosts. These findings not only provide evidence that the target-based HTS platform with the BRET-based synthetic ATG8-sensors is highly sensitive and versatile to identify autophagy modulators but also suggest that manipulation of fungal autophagy can suppress fungal pathogenicity.

## Discussion

Here, we report novel fungal autophagy inhibitors, EB and its analogs EO and PT, identified by target-based HTS using ATG8 synthetic sensors. Our screening method with the BRET-based *Bc*ATG8-sensor is specifically targeted at a key enzymatic reaction of the *Bc*ATG4-mediated processing of *Bc*ATG8 in the last step of autophagosome

formation. A single molecule BRET assay can increase sensitivity and decrease workload for HTS significantly. Furthermore, our target-based HTS platform is easily applicable to identify hit molecules for the regulation of autophagy biogenesis in different species because new BRET-sensors can be easily generated by replacing the open reading frame of *Bc*ATG8 with ATG8 genes of different species including vertebrates. Together with new BRET-sensors, our established protocol described here enables the identification of new autophagy modulators for different species.

Based on our results, the application of the new autophagy inhibitors identified suppressed germination, mycelial growth, and appressorium formation of *B. cinerea*, *M. oryzae*, *S. sclerotiorum*, and *M. fructicola*. Furthermore, autophagy inhibitors significantly reduced the virulence of these fungal pathogens in the corresponding hosts. Our study, together with previous research, veritably supports the notion that autophagy plays important roles in the development and pathogenicity of fungal pathogens.

EB has been shown to have anti-inflammatory and neuroprotective properties, and the clinical benefit of EB for human noise-induced hearing loss has been confirmed by oral administration to young adults in a phase 2 clinical trial[54]. Thus, the autophagy inhibitor EB and its analogs could be safe and effective antifungal compounds against important fungal pathogens. Therefore, autophagy is an excellent target for discovering chemical modulators that control fungal pathogen infection.

Due to the conservation of autophagy in eukaryotes, cross reactivity of fungal autophagy inhibitors in host cells might be of concern. Our HTS platform with the *Bc*ATG8-sensor allowed us to identify autophagy modulators for *B. cinerea*. This method needs to be further optimized for the identification of species-specific autophagy modulators that manipulate autophagy biogenesis only in fungal pathogens because host autophagy plays an important role in the regulation of the hypersensitive response programmed cell death (HR-PCD) during plant-microbe interactions[55,56]. In our experimental setup, while examining the growth inhibition of *B. cinerea* by EB-spraying on plant leaves, we have not detected a significant inhibition of plant autophagy activity (Supplementary Fig. 11a, b), indicating that the effect on the host autophagy inhibition by conventional application of EB appears to be minimal. This was further verified by multiple applications of EB to transgenic *N. benthamiana* expressing the *Arabidopsis At*ATG8a-sensor[33,34]. Although the transgenic plants were exposed to 300 μM EB for 1 week, the maturation of the *At*ATG8a-sensor was not affected (Supplementary Fig. 11a). We postulate that the physical barriers of plants that prevent water loss and protect pathogen invasion appear to prevent absorption of EB into plant cells, resulting in a minimal effect on host autophagy.

Selenium in the EB scaffold has been shown to make a covalent bond with a cysteine residue (Cys148) of thioredoxin reductase in *Aspergillus fumigatus*. This interaction inhibited the enzymatic activity, resulting in a disturbance of redox status in *A. fumigatus*[57]. The molecular mechanism for the effects of the lead compounds EB, EO, and PT on ATG4 could be similar to *A. fumigatus* thioredoxin reductase although the macromolecules targeted by EB and its analogs are different. To understand the mode of action of EB for the inhibition of *Bc*ATG4, we hypothesized that cysteine residues in *Bc*ATG4 could be targets of EB of which bindings could inhibit the *Bc*ATG4-mediated cleavage of *Bc*ATG8 since the thiol group in cysteine residues has been known to be a target of EB[40]. Therefore, the *Bc*ATG4 (C to A) mutants are resistant to EB treatment. For the roles of cysteine residues in ATG4, it has been revealed that *Hs*ATG4B C74 functioned as the catalytic cysteine at the active site and the *Hs*ATG4B C74A mutant showed loss of the enzymatic activity[58,59]. Among six cysteine residues in *Bc*ATG4, C158 is predicted to be the cysteine residue of the catalytic triad (C158, D332, and H334) at the active site of *Bc*ATG4 (Supplementary Fig. 12a). Consistent with this, our data show that the *Bc*ATG4

(C158A) mutant lost the enzymatic activity to cleave the *Bc*ATG8-sensor, indicating that C158 is the catalytic cysteine at the active site of *Bc*ATG4 (Supplementary Fig. 12b). Due to the absence of structural information of *Bc*ATG4, we performed in silico analysis to test the possibility that C158 could be a target for EB. First, we built a homology model of *Bc*ATG4 using RoseTTAFold[60]. Our model of *Bc*ATG4 was superimposed with the crystal structure of *Hs*ATG4B (PDB, 2cy7) except for two insertion parts in the N-terminal *Bc*ATG4 (Supplementary Fig. 12a). In particular, the active site of the *Bc*ATG4 model is structurally conserved with that of *Hs*ATG4B. Next, we performed docking simulation with the *Bc*ATG4 model and EB using AutoDock[61]. The docking simulation proposed that EB could access the active site of *Bc*ATG4 and interact with the catalytic C158. This prediction suggests that C158 in *Bc*ATG4 may be a target residue of EB (Supplementary Fig. 12c). To identify additional putative target cysteines in *Bc*ATG4, we compared the cleavage activity of wild type and different point mutants (C to A) under mock and EB treatment. In addition to C158A, the C233A mutant among five mutants showed the highest resistance to EB, indicating that C233 could be a putative target residue of EB in *Bc*ATG4 (Supplementary Fig. 12d). In addition, since the selenium-sulfur bond and the disulfide bond are reduced under a reducing condition[40,41], we confirmed that the inhibition of *Bc*ATG4 by EB was restored under DTT treatment (Fig. 1d). We have shown the direct interaction of EB with *Bc*ATG4 by ITC experiments (Fig. 1g). Taken together, in silico analysis and our experimental results suggest that EB can directly bind to putative target cysteines such as C158 and C233 in *Bc*ATG4. Binding of EB to the thiol group of C158 causes disturbance of the active site of *Bc*ATG4. It is of interest to experimentally prove whether the catalytic cysteine in *Bc*ATG4 is a target of EB. Another decoration of EB on C233 appears to induce allosteric inhibition. These modifications in *Bc*ATG4 cause the inhibition of cysteine protease activity, resulting in suppression of autophagy and attenuation of fungal virulence of *B. cinerea*.

EB has also been known as a glutathione mimetic[62]. Considering the function of a glutathione mimetic, we cannot rule out the possibility that EB can contribute to redox change, resulting in inhibition of the virulence of fungal pathogens. However, EO (possibly PT) with an inhibitory effect similar to EB does not have antioxidant activity to change the redox status in the cell[43]. Furthermore, antioxidant treatments with reduced L-Glutathione (GSH), N-Acetyl-L-cysteine (NAC), and ascorbic acid do not change the level of reactive oxygen species (ROS) in a developing germ tube and appressorium of *M. oryzae*, indicating that exogenous application of antioxidant reagents may be ineffective for normal development of infectious structures of *M. oryzae*[63]. More importantly, we confirmed autophagy inhibition from not only microscopy observation of abnormal autophagosomes (Fig. 2a, b) but also evaluation of PE-lipidation in GFP-*Bc*ATG8 (Fig. 2c) under EB treatment. Furthermore, we validated that the vegetative growth of wild type *S. sclerotiorum* was inhibited, while the genetic defect mutant, *Ssatg8*, was resistant to EB treatment (Supplementary Fig. 6d), indicating that the growth retardation of wild type could be due to the inhibition of autophagy by EB treatment. Based on our results, the development of abnormal germ tubes and the inhibition of *M. oryzae* appressorium formation under treatment with the EB compound (Fig. 3c and e) are not caused by the antioxidant activity of EB. Therefore, inhibition of the pathogenicity of fungal pathogens by EB, EO, and PT is attributed to suppression of autophagy rather than redox change in the cell. Additionally, attenuation of fungal pathogenicity is a common phenotype of the genetic defects of the core components of autophagy as observed in *Bcatg1*, *Bcatg4*, *Bcatg8*, *Mgatg1*, *Moatg4*, and *Moatg8* mutants[23,25–27,29,30]. Taken together, the new fungal ATG4 inhibitors identified in this study can inhibit the autophagy process, resulting in the prevention of fungal infection in host plants.

*Moatg8* and *Moatg4* mutants develop malfunctional appressoria, suggesting that autophagy may not be involved in the formation of appressoria but in the maturation process[23]. Our results show that autophagy inhibitors EB and its analogs suppress *M. oryzae* appressorium formation that has not been observed in *Moatg* mutants. Recently, the *MoPmk1-MoHOX7* signal cascade has been found to control transcription of autophagy genes, indicating that autophagy is downstream of the *MoPmk1-MoHOX7* signaling axis in *M. oryzae*[64]. In addition, appressorium formation is abolished in the *pmk1* mutant[65] as shown in the treatment of our lead compounds EB, EO and PT. Thus, we postulate that EB, EO, and PT can target a component in the *MoPmk1-MoHOX7*-autophagy signaling cascade in addition to *Mo*ATG4 during the formation of the appressorium of *M. oryzae*. Nevertheless, the synergic effect on the developmental constraints of appressorium formation and maturation by the lead compounds EB and its analogs results in inhibition of *M. oryzae* infection of rice.

In summary, our findings indicate not only that autophagy is an excellent antifungal target, but also that the new autophagy modulators inhibit the pathogenicity of important fungal pathogens in *Ascomycota*. The ATG8-sensor based screening strategy described here with a larger collection of chemical compounds should identify additional novel autophagy modulators that enable the prevention of infection of devastating fungal pathogens.

## Methods

### Plant materials and growth conditions

*Solanum lycopersicum* cv. Moneymaker seeds were germinated at 25 °C in the dark for 5 days and then transplanted to a 50 plug tray. *N. benthamiana* seeds were sown in 200 plug form trays filled with commercial soil and the seedlings were transplanted to pots. Both species were grown in a controlled growth chamber at ~24 °C and 60% relative humidity under a 16 h-light/8 h-dark period. Surface sterilized seeds of *Oryza sativa* cv. Nakdong were germinated in distilled water in the dark and seedlings were transplanted to rice nursery soil. Rice plants were grown in a growth chamber at 28 °C and 80% humidity with a 16 h-light/8 h-dark photoperiod.

### Fungal strains and culture conditions

*B. cinerea* strain B05.10 was used as a wild type strain in this study. All strains were maintained on potato dextrose agar (PDA) (Difco) in a growth chamber at 23 °C. *M. oryzae*, a wild type KJ201, was provided by the Center for Fungal Genetic Resources (CFGR; http://cfgr.snu.ac.kr). *M. oryzae* was grown on V8 juice agar medium [8% V8 juice (v/v), 1.5% agar powder (w/v), pH 6.7] at 25 °C under a continuous fluorescent light and harvested after a week. *M. fructicola* strain Mf13–81 was maintained on V-8 juice agar at RT and mycelial plugs (5 mm) for assays were collected from 7–10 day-old cultures. Wild type of *S. sclerotiorum* strain 1980 was maintained on PDA at 24 °C. For the *Ssatg8* strain, it was maintained on PDA containing hygromycin B (100 μg/mL). To examine inhibition of the vegetative growth of both wild type and *Ssatg8* under EB treatment, each fungal strain was grown on PDA containing either 0.1% DMSO or 5 μM EB. Whereas wild type was grown under the EB treatment for 40 hpi, the growth inhibition of *Ssatg8* was measured at two time points (40 and 65 hpi) due to the different growth rate of *Ssatg8*, compared with that of wild type. 40 hpi is the same incubation time of both wild type and *Ssatg8* under the EB treatment and 65 hpi is the time point for the *Ssatg8* growth with a similar growth size as wild type. 0.1% DMSO was a mock treatment.

### Molecular cloning

*Bc*ATG4 and *Mo*ATG4 coding sequences were PCR amplified and cloned into the *BamH*I-*Not*I site in the pET-28a bacterial expression vector. *Bc*ATG8 and *Mo*ATG8 coding sequences without the stop codon were PCR amplified and cloned into the *BamH*I-*Sal*I site in pET28-Citrine-ShR vector[34] resulting in C-*Bc*ATG8-ShR and C-*Mo*ATG8-ShR plasmids. To make the GFP-*Bc*ATG8 construct, the full-length ORF of *Bc*ATG8 was amplified with primer pairs gF/gR-*Bc*ATG8 from *B. cinerea*

B05.10 cDNA and cloned downstream of GFP driven by the constitutive *Olic* in the vector pNAH-OGG at *Not*I site. Open reading frames of inserts in all plasmids were confirmed by sequencing. All primer information used for experiments will be provided upon request.

## Protein expression and purification

The N terminus 6 × His expression constructs were transformed into *Escherichia coli* strain BL21. Protein expression was induced with 1 mM IPTG at $OD_{600}$ of 0.6-0.8 and the culture was incubated at 18 °C overnight. Soluble proteins were loaded in a column chromatography cartridge packed with cobalt resin (ThermoScientific), washed with a washing buffer (50 mM sodium phosphate, 300 mM sodium chloride, 10 mM imidazole; pH 7.4), and eluted with an elution buffer (50 mM sodium phosphate, 300 mM sodium chloride, 150 mM imidazole; pH 7.4). Affinity-purified recombinant proteins were dialyzed against a PBS (pH 7.2) buffer at 4 °C overnight and the dialyzed proteins were concentrated using either 10 K or 30 K centrifugal filter units (Merck). Concentrated *Bc*ATG4 and *Bc*ATG8 were gel filtrated using a Superdex 200 and Superdex 75 (GE healthcare) on an AKTA explorer system, respectively. Purified proteins were quantified using BSA (Thermo-Scientific) as a standard and confirmed by SDS-PAGE after Coomassie brilliant blue staining (Bio-rad).

## Chemical library screening and BRET measurement

In typical HTS, the ability to identify hit molecules is largely dependent on suitability and quality of assay platforms. Z-factor is a simple statistical parameter to evaluate the quality of particular HTS assays. Z-factor is defined as a screening window coefficient including the assay signal dynamic range and the data variation of sample measurements. Thus, the z-factor is a useful and widely applied factor able to determine the quality of HTS[66]. Z-factor is defined as the following equation:

$$Z - factor = 1 - [3(\sigma_p + \sigma_n)/|\mu_p - \mu_n|]$$

$\sigma_p$: standard deviation of positive control.
$\sigma_n$: standard deviation of negative control.
$\mu_p$: means of positive control.
$\mu_n$: means of negative control.

In brief, chemical library screening[35] was optimized. Approximately 100 ng of purified *Bc*ATG4 and 10 μM of each chemical were incubated at RT for 10 min and then 200 ng of the *Bc*ATG8-sensor added in a 96-well plate. 100 μM solution of coelenterazine (CLZ) was prepared by dilution with 100% ethanol and protected with aluminum foil from light because CLZ is light sensitive. BRET measurement using a microplate reader (TECAN) was recorded after automatic injection of 20 μL of 100 μM of CLZ. 460 nm and 540 nm filters were used to measure blue luminescence and yellow fluorescence, respectively. For testing the detection sensitivity of HTS with the *Bc*ATG8-sensor, 100 ng of the *Bc*ATG8-sensor and 200 ng of *Bc*ATG4 were serial diluted in a 96-well plate and then incubated for 10 min. After the incubation, the rest of the procedure including BRET measurement was performed as explained before.

## In vitro cleavage assay and western blot

Approximately 100 ng of the purified *Bc*ATG4 cysteine protease and the point mutants with 10 μM of chemicals (EB, EO, and PT) were incubated at RT for 10 min. 200 ng of the *Bc*ATG8-sensor was added and then incubated for an additional 10 min. Total reaction volume was 20 μL. *Mo*ATG4 and the *Mo*ATG8-sensor were tested in the same way. The total proteins were boiled for 5 min at 95 °C with 2X Laemmli buffer (Bio-rad) and separated on Mini-protean TGX precast gels 4–20% Resolving Gel (Bio-Rad) using Tris-glycine SDS buffer at 50 mA for 45 min. After SDS-PAGE, the proteins were transferred to a 0.45 μM PVDF (polyvinylidene difluoride) membrane (Bio-Rad) using the Trans-Blot Turbo Blotting System (Bio-Rad). After transferring the proteins onto a PVDF membrane, immunoblot analyses were performed using monoclonal antibodies against RLUC (ShR) (1:10000, Millipore Sigma) and anti-mouse secondary antibody (1:10000, ThermoScientific) following by ECL™ Prime Western Blotting System (GE healthcare). To estimate relative enzyme activity, ImageJ (National Institutes of Health) and Prism 7.0 (GraphPad) were used for quantification.

## Isothermal titration calorimetry (ITC)

ITC experiments were carried out at 20 ˚C using an Affinity ITC Titration (TA instruments). For the interaction between a protein and a chemical, the calorimetric cell was filled with a chemical and titrated with each ATG4 in the syringe. A single injection of 10 μL of ATG4 was consecutively injected up to 30 times. Injections were made at 300 s or 600 s intervals with a stirring speed of 125 rpm. All raw titration data were integrated and fitted to a multiple site-binding model using the TA instrument analysis software.

## Generation of *B. cinerea* transgenic strains expressing the *GFP-BcATG8* cassette and monitoring autophagy in vivo

Linearized fusion of the *GFP-BcATG8* fragment was amplified by primer pairs Pmt-F/R and transformed into *B. cinerea* B05.10 protoplasts[67,68]. The resulting transformants with hygromycin resistance were screened by PCR and the GFP signal was observed by fluorescence microscopy. All primer information used for experiments will be provided upon request. For monitoring autophagy in the cultures of transgenic *B. cinerea* expressing the *GFP-BcATG8* cassette, the strain was maintained on PDA containing hygromycin B (100 μg/mL) at 23 °C in a growth chamber for 2 weeks.

For a confocal microscopy, conidia of *B. cinerea* expressing the *GFP-BcATG8* cassette were collected from PDA and then cultured onto black culture dish (Electron Microscopy Science) in potato dextrose broth (PDB) for 12 hrs at 25 °C. After careful removing of PDB, the conidia were washed thoroughly with distilled water and then, 0.1% low gelling agarose (Sigma) with 1 μM of concanamycin A (Santa Cruz Biotechnology) and either 1% DMSO for mock treatment or 20 μM EB were applied for 12 hrs. Images were generated in a ZEISS LSM980 AiryScan super-resolution microscope system using Axio observer Z1 inverted microscope with 40X/1.2 NA C-Apochromat water immersion objective for Fig. 2a. or 63X/1.4 NA plan-Apochromat oil immersion objective for the Airyscan superresolution images[69] in Fig. 2b. For the superresolution images, images were scanned with 0.17 μm interval followed by 2D Airyscan processing. Images of about 1 μm section were superimposed. About 1–5% of the 488 nm argon laser was used to excite GFP. Images were resized and cropped using ImageJ (NIH). Five images were generated in each experiment and the experiments were repeated three times.

For a pulse-chase experiment, $1 \times 10^4$ conidia/mL of the transgenic *B. cinerea* strain were freshly collected and seeded in 25% organic grape juice in a well of a 6-well plate (Corning). After germination of spores at 23 °C overnight, the fungal lawn was washed with sterilized water three times. Either 50 or 100 μM of EB was treated in a well containing 2 mL of sterilized water. 1% DMSO was used as a mock treatment. The plate was placed at 23 °C for 8 hrs and then, the fungal lawn was collected after brief washing with sterilized water. 120 μL of 2X Laemmli buffer (Bio-Rad) was added to fungal samples and then, samples were vortexed for 10 min. Samples were boiled for 5 min and supernatants were collected after centrifugation at 12000 g for 5 min. Each sample was separated in a 6 M urea 12% SDS PAGE gel at 125 V for 2 hrs followed by immunoblotting with α-GFP (Invitrogen) or α-H3 antibody (Agrisera).

## qRT-PCR

The sample preparation for RNA extraction of wild type *B. cinerea* was performed as described in the sample preparation for the microscopic experiment of GFP-*Bc*ATG8. RNA was extracted from the collected

mycelium by RNeasy Plant Mini Kit (Qiagen). One μg of total RNA was subjected for reverse transcription reaction using ProtoScript® II First Strand cDNA Synthesis Kit (NEB). For the qPCR, the reaction mixture containing cDNA template, primers, and SYBR Green PCR Master Mix (Invitrogen) was run in a CFX Connect Real-Time PCR Detection System (Bio-rad). All raw gene expression data were integrated and normalized using the CFX Maestro Software (Bio-rad). The primer information used for experiments will be provided upon request.

#### Quantification of conidial germination and appressorium formation

Conidia were harvested from 10 day-old cultures grown on PDA (*B. cinerea*) and 7 day-old cultures on V8 juice agar (*M. oryzae*) plates with sterilized distilled water and filtered through two layers of Miracloth (Merck). Conidial germination and appressorium formation were assessed on hydrophobic microscope coverslips (SUPERIOR). A droplet of conidial suspension adjusted to $3 \times 10^4$ conidia/mL was applied to a coverslip and incubated in a humid box at 23 °C, with three technical replicates per treatment. Frequencies of germination and appressorium formation were measured by counting at least one hundred conidia per replicate under a light microscope. Stock solutions of chemicals were prepared in DMSO of which concentration was 1% after the dilution. Ten μL of either a drug stock or DMSO alone (control) was added in 990 μL of conidial suspension, mixed well and 30 μL of each mixture was dropped on the coverslip with three technical replicates. The *M. oryzae* conidial staining with propidium iodide was performed to investigate the membrane integrity (Invitrogen). An aqueous propidium iodide solution of 100 μg/mL in 1 × PBS was used to stain conidia. Freshly collected conidia were incubated with either 1% DMSO or 20 μM EB for 4 h and then washed with 1 × PBS. The conidial membranes were stained with the working propidium iodide solution. Images were acquired by Leica TCS SP8 STED confocal microscope (Leica) under 488 nm WhiteLight laser excitation and 617 nm emission maxima. 1% DMSO was a mock control.

#### Infection test

For the protective activity tests, detached four *N. benthamiana* leaves were placed in a square dish (245 x 245 x 28 mm. SPL) and 300, 500, or 700 μM of EB dissolved in 10% acetone was applied on the leaves to run-off by 15 mL spray bottles. After the chemical dried on *N. benthamiana* leaves, ten μL of conidial suspension ($1 \times 10^4$ conidia/mL in 25% organic grape juice) was inoculated, incubated for 3 days and then photographed. For the curative activity tests, inoculated leaves had been incubated for 36 h and then 300 μM of EB was applied once per day. After 3 and 4 dpi, the disease lesion was measured. Ten percent acetone was sprayed as a control. All inoculated leaves were placed in humid boxes under high-relative humidity conditions at 24 °C with 16 h-light/8 h-dark period. 4 week-old rice plants were used for spray inoculation of *M. oryzae*. Conidial suspension ($5 \times 10^4$ conidia/mL in 250 ppm Tween 20) containing 100 μM of EB was sprayed and the inoculated rice plants were incubated in the dew chamber for a day. The inoculated plants were then transferred to a growth chamber and incubated for 6 days. To quantify disease symptoms, a numerical scoring system based on the previous study was used[70]. The disease severity was rated on a scale from 0 to 5, with 0 indicating no necrotic or chlorotic flecks on leaves. Inoculation of conidia suspension droplets of transgenic *B. cinerea* was performed following a slightly modified method from the protocol described[71]. In brief, ten μL of a conidial suspension ($1 \times 10^4$ conidia/mL in 25% organic grape juice) contained each lead compound of which final concentration was 50 μM. Single droplet was carefully loaded on tomato leaves. After drying the droplet on the leaves, the inoculated leaves were incubated in humid boxes for 3 days. The droplets with 1% DMSO were used as a control. For the low dose test, conidial suspension ($1 \times 10^4$ conidia/mL in 25% organic grape juice) contained EB of which final concentrations were 5 or 10 μM was used for inoculation. A single droplet was carefully loaded on *N. benthamiana* leaves. The inoculated leaves were incubated in humid boxes for 3 days. The droplets with 1% DMSO were used as a control. For *S. sclerotiorum* infection, Three-mm-diameter agar plugs containing actively grown hyphal tips from 3 day-old colonies of *S. sclerotiorum* on PDA containing 50 μM of EB were inoculated onto detached plant leaves. Inoculated leaves were placed on moistened sterile filter paper in glass Petri dishes and incubated at 25 °C under high humidity conditions. 1% DMSO was used as a control. The lesion diameter was recorded at 48 hpi.

#### Monitoring host autophagy

To investigate host autophagy responses, the one-month-old transgenic *N. benthamiana* expressing the *Arabidopsis* BRET-sensor (Citrine-*At*ATG8a-ShR) was subjected for EB-spraying application.

300 μM of EB was applied once per 2 days for 3 times. After 7 days, samples were collected. The transgenic *Arabidopsis* expressing GFP-*At*ATG8 was also used to monitor host autophagy under the exogenous application of EB. Samples were collected at the indicated time and frozen in liquid nitrogen and homogenized in protein extraction buffer [GTEN (10% glycerol, 25 mM Tris pH 7.5, 1 mM EDTA, 150 mM NaCl), 2% w/v PVPP, 10 mM DTT, 1X protease inhibitor cocktail (Roche) and 0.1% Tween 20]. Resuspended samples were centrifuged at 13000 rpm for 5 mins at 4 °C. Supernatants after centrifugation were boiled at 95 °C in 2X Laemmli buffer (Bio-rad) for 10 min. Extracted plant proteins were analyzed to monitor autophagy activities by western blot.

#### Homology modeling and docking simulation with *Bc*ATG4 and EB

The entire amino acid sequence of *Bc*ATG4 was used to build a homology model using RoseTTAFold[60]. The homology model was superimposed with human ATG4B (PDB, 2cy7) by the Swiss PDB viewer. For docking simulation, the homology model of *Bc*ATG4 was used for receptor preparation and the canonical SMILES of EB was used for ligand preparation. Docking simulation was performed using AutoDock[61] provided by a web-based program, SeamDock (https://bioserv.rpbs.univ-paris-diderot.fr/services/SeamDock/). To predict if EB could access the active site of *Bc*ATG4, the docking simulation was performed with the active site region of the *Bc*ATG4 homology model.

#### Reporting summary

Further information on research design is available in the Nature Portfolio Reporting Summary linked to this article.

## Data availability

All data are available in the main text or the supplementary information and the source data files. Source data are provided with this paper.

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

## Acknowledgements

The authors thank NIH-NCATS for providing the chemical collection approved for clinical trials and thank Dr. Thomas Boothby for sharing the microscope to generate super-resolution images. This work was supported by grants from US the National Science Foundation (MCB-EAGER-1355459, MCB-EAGER-1549580 and IOS-1339185) to S.P.D-K and the Korea National Research Foundation (NRF) grant funded by the Korea government (MSIT) (2018R1A5A1023599, SRC) to Y.-H.L., D.C., and E.P. An NSF award (EPS-1655726 for E.P.'s start-up), INBRE thematic grant (IDeA from NIGMS, NIH, 2P20GM103432) to E.P., and NSF IOS-2126256 to E.P. and J.W. also supported this work.

## Author contributions

S.P.D.-K., J.W., E.P., and M.D. conceptualized the project, designed experiments, and performed analyses; J.W., S.J., S.K., Y.L., H.C., T.R., H.Z., C.C., and E.P. performed experiments and analyzed the data; J.W., S.J., E.P., and S.P.D-K. drafted the manuscript and D.K., K.-N. K., R.M.B., Y.-H.L., and D.C. contributed to reviewing and editing the manuscript.

## Competing interests

J.W., E.P. at University of Wyoming and S.P.D.-K. at University of California, Davis have filed a provisional patent application encompassing aspects of this work. The remaining authors declare no competing interests.
