## [Peer Review File · Nature Communications]

Attenuation of phytofungal pathogenicity of Ascomycota by autophagy modulatorsReviewer #1 (Remarks to the Author):

Autophagy plays important roles in pathogenicity of various fungal pathogens. In this manuscript, authors describe BRET-based HTS strategy to identify compounds that inhibit fungal ATG4 cysteine protease-mediated cleavage of ATG8. EB and its analogs were identified as inhibitors of fungal pathogens, and can inhibit spore germination, hyphal development and appressorium formation. Treatment with EB and its analogs significantly reduced fungal pathogenicity. The findings may help to develop the next generation of antifungal compounds by targeting autophagy in important fungal pathogens. However, a major revision should be performed and some important concerns should be explained before the manuscript can be accepted for publication.

Major Comments:

1. Authors claimed that they developed a BRET-based HTS strategy to identify compounds that inhibit fungal ATG4 cysteine protease-mediated cleavage of ATG8. But in the manuscript, I cannot find any description about the improvement of this strategy. It just like a normal experiment method? Did it show advances in sensitivity or detection range?
2. In my opinion, the concentration of EB in most of experiments is high (usually higher than 10 μM), therefore, the potential of ATG4 as fungicide target may be not fully explored. Or in other way, I think the compound should be optimized.
3. The aim of selecting *B. cinerea* and *M. oryzae* for testing the autophagy induced by EB should be explained?
4. In in vivo test, the conidial suspension of *B. cinerea* should be used in the infection test. In addition, the protective and curative activity of EB and its analogs should be determined.
5. Autophagy is a complex process including induction, vesicle nucleation, extension, mature and cracking. The stage at which the drug acts should be determined. Furthermore, I suggest authors use RNAseq and qRT-PCR to identify the influence of EB to the expression level of autophagy genes in vivo to avoid off-target toxicity.
6. TEM should be used to observe the structure of autophagosomes and the formation process.
7. The mechanism of EB for controlling fungi disease should be studied in more depth.

Minor Comments:

1. The manuscript mainly focused on ATG4 and ATG8. But in the third paragraph of Introduction part, there were some description of ATG1, which may interfere with readers' understanding of the manuscript. I suggest authors reconsider the importance of these description.
2. In the Introduction, the fungicide-resistance for *B. cinerea* and *M. oryzae* is not exactly true. For *B. cinerea*, the fungicide-resistance is serious.
3. The binding affinity between EB and ATG4 should be determined by another technology such as MST or SPR.
4. EB showed limited chemical size, did authors consider their toxicity? More importantly, I suggest authors introduce the limitations about this manuscript in the Discussion part.

Reviewer #2 (Remarks to the Author):

The authors use BRET-based HTS strategy to identify inhibitors of autophagy, then they validated the inhibition of EB and its analogs on spore germination, hyphal development, appressorium formation and pathogenicity of four different Ascomycota pathogens. The manuscript is well written and the authors did an impressive amount of work to validate their findings.

Minor comment:

the authors have gotten the Arabidopsis transgenic plants expressing the synthetic substrate C-AtAtg8a-ShR in reference 33 and 34, why the authors did not use these plants to analysis cross reactivity of autophagy inhibitors in host cells for longer time? EB perhaps has cross inhibiting reactivity in one week or longer time.

Response to Reviewer's Comments

We appreciate the reviewers for their efforts and constructive comments on our manuscript. We have included new data based on the reviewer's comments in Figure 2b and Supplemental Figures 1b, 1e, 4a, 5b, 10c, and 11a, and 12. The revised text in the revised manuscript is highlighted in yellow. Below is our point-by-point response to reviewers' comments.

Response to Reviewer #1:

Autophagy plays important roles in pathogenicity of various fungal pathogens. In this manuscript, authors describe BRET-based HTS strategy to identify compounds that inhibit fungal ATG4 cysteine protease-mediated cleavage of ATG8. EB and its analogs were identified as inhibitors of fungal pathogens, and can inhibit spore germination, hyphal development and appressorium formation. Treatment with EB and its analogs significantly reduced fungal pathogenicity. The findings may help to develop the next generation of antifungal compounds by targeting autophagy in important fungal pathogens. However, a major revision should be performed and some important concerns should be explained before the manuscript can be accepted for publication.

We thank the reviewer for the positive comments that our findings will help to develop novel antifungal compounds by targeting fungal autophagy.

Major Comments:

1. Authors claimed that they developed a BRET-based HTS strategy to identify compounds that inhibit fungal ATG4 cysteine protease-mediated cleavage of ATG8. But in the manuscript, I cannot find any description about the improvement of this strategy. It just like a normal experiment method? Did it show advances in sensitivity or detection range?

Our approach is advanced in two aspects:

1. Utilizing BRET-based autophagy target screening for the discovery of potential new antifungal or fungicidal agents is novel and has not been used for this purpose previously.
2. Our screening method is specifically targeted to a key molecular activity of the *Bc*ATG4-mediated processing of *Bc*ATG8 that is the last step in the autophagosome formation process. Single molecule BRET assay can increase sensitivity and decrease workload for high throughput screening significantly. Additionally, our target-based HTS platform enables other researchers to identify new autophagy modulators by simply replacing the *Bc*ATG8 open reading frame with other ATG8 open reading frames from different species including vertebrates following our established protocol described in this paper.

To further provide evidence for the novelty and sensitivity of our HTS platform, we determined the detection range of EB in our HTS assay. Our HTS screening could distinguish the significantly different BRET ratio between the uncleaved *Bc*ATG8-sensor and the *Bc*ATG8-sensor cleaved by *Bc*ATG4 at 177.3 pM of the sensor. For the detection capability of the inhibition of *Bc*ATG4 by EB, ~0.3 μ M EB showed increased BRET ratio, compared to mock treatment. The detection range with ~300 nM of EB indicates our HTS is highly sensitive, compared to other research based on cell-based screening (Konstantinidis, et al., 2018; Forveille et al., 2021).

We have shown these results in new Supplementary Fig. 1b and 1e in the revised manuscript. See corresponding text in the results section, lines 124-129 and 134-137.

References:

- Konstantinidis, G., Sievers, S., Wu, YW. (2018). Identification of novel autophagy inhibitors via cell-based high-content screening. *Methods Mol. Biol.* 1854:187-195.
- Forveille, S., Leduc, M., Sauvat, A., Cerrato, G., Kroemer, G., Kepp, O. (2021) High throughput screening for autophagy. *Methods Cell Biol.* 165:89-101.

2. In my opinion, the concentration of EB in most of experiments is high (usually higher than 10 μ M), therefore, the potential of ATG4 as fungicide target may be not fully explored. Or in other way, I think the compound should be optimized.

Based on the reviewer's suggestion, we evaluated a low-dose effect on inhibition of *B. cinerea* infection on live host. For this, we prepared conidia suspension (as suggested by the reviewer in comment #4) and mixed with low-dose EB (5 or 10 μ M) and spotted onto *Nicotiana benthamiana* leaves. In this assay, 10 μ M EB is sufficient to suppress the disease progression of *B. cinerea* on *N. benthamiana* leaves, indicating a low-dose of EB is effective in suppressing the infection. We have shown these results in new Supplementary Fig. 5b and added corresponding text (lines 227-231) in the revised manuscript.

For our screen, we used the *Bc*ATG4-mediated cleavage of *Bc*ATG8 because of the evidence from *Bcatg4* and *Bcatg8* mutants. The *Bcatg4* and *Bcatg8* mutants show loss of pathogenicity (Ren et al., 2018; Liu et al., 2019), indicating that the enzymatic process of the maturation of *Bc*ATG8 is an excellent target to identify antifungal compounds. The concentration of EB that we have used is reasonable when compared to the published reports: Vicedo et al., used an antifungal compound of hexanoic acid to test the growth inhibition of *B. cinerea* up to 4 mM. Commercial fungicides of Rovral and Celest have been tested against *B. cinerea* ranging from 0.3 μ M to 12 mM (Kim et al., 2016). In addition, the antifungal activity of Difenoconazole has been evaluated at 197 μ M against *B. cinerea* (Mamiev et al., 2020). Given different fungicides have different modes of action and different efficacies against a target fungus, we have used different concentrations of EB to evaluate antifungal activity in different experimental conditions. In *in vitro* conditions, target-based HTS and the cleavage assay of the *Bc*ATG4-mediated cleavage of *Bc*ATG8, 10 μ M is sufficient to monitor inhibition of the enzymatic reaction since the detection capability for EB is 0.3 μ M in our HTS (Supplementary Fig. 1b and 1e). For *in vivo* experiments, we tested different concentrations to evaluate the suppression of fungal growth (Fig. 2, 3, and 4; Supplementary Fig. 5 and 8). Taken together, compared with other published studies, the concentrations we used are reasonable. Furthermore, one could modify the side group of EB in the future to enhance its activity; we believe that this is beyond the scope of this manuscript.

References:

- Mamiev et al., (2020). Effectiveness of different fungicides in controlling botrytis grey mould of tomato. *IOP Conf. Series: Earth Env Sci.* 614:012112.
- Kim et al., (2016). Effectiveness of different classes of fungicides on *Botrytis cinerea* causing gray mold on fruit and vegetables. *Plant Pathol. J.* 32:570-574.
- Vicedo et al., (2009). Hexanoic acid-induced resistance against *Botrytis cinerea* in tomato plants. *Mol. Plant-Microbe Interact.* 22:1455-1465.
- Liu et al., (2019). Involvement of the cysteine protease *Bc*Atg4 in development and virulence of *Botrytis cinerea*. *Curr Genet* 65:293-300.
- Ren et al., (2018). The autophagy gene *Bc*ATG8 regulates the vegetative differentiation and pathogenicity of *Botrytis cinerea*. *Appl Environ Microbiol* 84:e02455-17.

3. The aim of selecting *B. cinerea* and *M. oryzae* for testing the autophagy induced by EB should be explained?

We thank the reviewer for this suggestion. We have included the explanation in the introduction and results section. See lines 81-94 and 114-119.

4. In in vivo test, the conidial suspension of *B. cinerea* should be used in the infection test. In addition, the protective and curative activity of EB and its analogs should be determined.

We thank the reviewer for the suggestion. In the previous version of the manuscript, we did use conidial suspension of *B. cinerea* for all our leaf infection tests, either dropping on one spot or spraying the entire leaf (Fig. 4c-d and Supplementary Fig. 4b). We revised the manuscript to make this clear (see lines 287-289 and 291-296).

When we designed the project, our main focus was *B. cinerea* requiring more protective activity by fungal inhibitors for post-harvest storage. However, we valued the reviewer's suggestion to test the curative activity of EB to inhibit the disease progression. Therefore, we examined the curative activity of EB in the live plant tissues. Prior to spray-application of EB, the conidia suspension of *B. cinerea* was inoculated and the disease symptom was incurred on the host tissues at 36 hpi. Application of EB after disease development slowed down the disease progression compared to the control. These results are shown in new Supplementary Fig. 10c. Corresponding text is in lines 293-299.

5. Autophagy is a complex process including induction, vesicle nucleation, extension, mature and cracking. The stage at which the drug acts should be determined. Furthermore, I suggest authors use RNAseq and qRT-PCR to identify the influence of EB to the expression level of autophagy genes in vivo to avoid off-target toxicity.

As pointed out by the reviewer and mentioned in the manuscript, autophagy is a complex process to maintain homeostasis in the cell. The role of each core component has been well characterized in the autophagy process. Furthermore, the functions of autophagy core components are well conserved across eukaryotes. According to many published literatures including from our group, ATG8 is the substrate of ATG4. The maturation of ATG8 by ATG4 enables the matured ATG8 to bind with phosphatidylethanolamine (PE) that is a membrane lipid. The lipidated ATG8 is then delivered to phagophore to supply membrane lipids for the elongation of phagophore. All ATG8 and all ATG4 play a role in the last step of autophagosome biogenesis (completion of autophagosome formation) (Marshall and Vierstra, 2018; Mizushima et al., 2011; Kabeya et al., 2004, and Kirisako et al., 2000).

As suggested by the reviewer, we estimated transcript levels of known autophagy core components such as *BcATG1*, *BcATG3*, *BcATG4*, *BcATG6*, *BcATG7*, and *BcATG8* by qRT-PCR. Our result showed that the transcription of the autophagy core components was not affected by EB treatment, indicating EB did not affect the transcription of core autophagy components. These results are shown in new Supplementary Fig. 4a and corresponding text is added in lines 187-191.

We don't think the EB-mediated autophagy inhibition in the fungal cells is due to change in transcription of the autophagy core components because the fungal samples collected for qRT-PCR is the same treatment condition used for the super-resolution microscopy. We observed abnormal formation of autophagosomes under the super-resolution microscopy (see new Fig. 2b). Please see more details on this in comment #6 below. Taken together, the formation of abnormal autophagosomes in the fungal cells is due to the inhibition of the enzymatic activity of *BcATG4* under EB treatment. In the revised manuscript, we have included these results in Fig. 2b and Supplementary Fig. 4a. The corresponding text is in 180-187.

References:

- Marshall and Vierstra, (2018). Autophagy: The master of bulk and selective recycling. *Annu Rev Plant Biol* 69:173-208.
- Mizushima et al., (2011). The role of Atg proteins in autophagosome formation. *Annu Rev Cell Dev Biol.* 27:107-132.
- Kabeya et al., (2004). LC3, GABARAP, and GATE-16 localize to autophagosomal membrane depending on form-II formation. *J. Cell Sci.* 117:2805-12.
- Kirisako et al., (2000). The reversible modification regulates the membrane-binding state of Apg8/Aut7 essential for autophagy and the cytoplasm to vacuole targeting pathway. *J. Cell Biol.* 151:263-75.

6. TEM should be used to observe the structure of autophagosomes and the formation process.

With wild-type *B. cinerea*, autophagy biogenesis has been well studied by a variety of experiments (Liu et al., 2019; Ren et al., 2018). TEM is indeed good to visualize the entire process of the autophagosome formation. We can observe from the partial phagophore to the complete double membranous autophagosome. However, we reason that TEM is not the best approach to address inhibition of autophagy under the drug treatment due to the unsynchronous effect of the drug inhibition between cell to cell. TEM is good to generate qualitative data rather than quantitative data. Therefore, we have used gold standard cell and biochemical approach to show inhibition of autophagy by EB treatment in the quantitative manner; GFP-*BcATG8* imaging that marks autophagosomes (Fig. 2a-b) and GFP-*BcATG8* lipidation analyzed by 6M Urea SDS-PAGE (Fig. 2c). These are well accepted quantitative approaches (Klionsky et al., 2021).

Because of reviewer's suggestion in comment #5, we were also interested in determining the stage of EB effect in autophagy biogenesis. Therefore, we further analyzed EB effect in autophagy biogenesis using a super-resolution confocal microscopy (Wu and Hammer, 2021) in live fungal cells expressing GFP-*BcATG8* under EB treatment, compared to mock treatment. With the super-resolution microscopy images, we confirmed well-described phenomena for autophagy inhibition - we often observed not only incomplete autophagosomes in the cytoplasm but also less accumulation of autolysosomes in the vacuole under EB treatment whilst we seldom observed the intracellular phenomena under mock treatment. Since the *BcATG4*-mediated processing of *BcATG8* is important to deliver a membrane lipid of PE to phagophores, the inhibition of the process causes incompleteness of autophagosome that cannot fuse to the vacuole, resulting in less accumulation of autolysosomes under EB and concanamycin A treatment. Thus, the observed phenomena of both abnormal formation of autophagosomes and less accumulation of autolysosomes are attributed to the inhibition of the *BcATG4*-mediated cleavage of *BcATG8* by EB in the last step of the autophagosome formation. We described these new results in the new Fig. 2b and included corresponding text in lines 180-187.

References:

- Klionsky et al., (2021). Guidelines for the use and interpretation of assays for monitoring autophagy (4th edition). *Autophagy* 17:1-382.
- Wu and Hammer, (2021). ZEISS Airyscan: Optimizing usage for fast, gentle, and super-resolution imaging. *Methods Mol. Biol.* 2304:111-130.
- Liu et al., (2019). Involvement of the cysteine protease *BcAtg4* in development and virulence of *Botrytis cinerea*. *Curr Genet* 65:293-300.
- Ren et al., (2018). The autophagy gene *BcATG8* regulates the vegetative differentiation and pathogenicity of *Botrytis cinerea*. *Appl Environ Microbiol.* 84:e02455-17.

7. The mechanism of EB for controlling fungi disease should be studied in more depth.

It is well-established that the *BcATG4*-mediated cleavage of *BcATG8* in the fungal autophagy is one of the most important cellular processes in development and pathogenicity because *Bcatg4* and *Bcatg8* mutants

show inhibition of fungal autophagy biogenesis, resulting in the growth inhibition and the loss of pathogenicity of *B. cinerea* (Ren et al., 2018; Liu et al., 2019).

Findings described in our manuscript clearly demonstrate that EB mode of action is through *BcATG4* and autophagy. Based on our results shown under EB treatment, the inhibition of activity of the cysteine protease, *BcATG4* (Fig. 1d, e, and f) causes to suppress autophagy of *B. cinerea* (Fig. 2a-c), resulting in the developmental arrests (Fig. 3, supplementary Fig. 5a, 5c, 5d, and supplementary Fig. 8) and the loss of pathogenicity (Fig. 4, supplementary Fig. 5b, and Supplementary Fig. 10). Based on the data shown in the manuscript, we propose that the loss of pathogenicity of fungal pathogens results from developmental perturbation induced by the autophagy inhibition under the treatment with EB.

To understand the mode of action of EB for the inhibition of *BcATG4*, we hypothesized that catalytic cysteine (C158) could be target of EB. Therefore, *BcATG4* (C158A) mutant could be resistant to EB treatment since the thiol group in cysteine residues has been known as a target of EB (Terentis et al., 2010). For the roles of cysteine residues in ATG4s, it has been revealed that *HsATG4B* C74 functions as the catalytic cysteine in the active site and the *HsATG4B* C74A mutant showed loss of the enzymatic activity (Shu et al., 2010). Among six cysteine residues in *BcATG4*, C158 is predicted as the cysteine residue of the catalytic triad (C158, D332, and H334) in the active site of *BcATG4* (Supplementary Fig. 12a). Consistent with this, our data shows that the *BcATG4* (C158A) mutant lost the enzymatic activity to cleave ATG8 sensor, indicating that C158 is the catalytic site in *BcATG4* (Supplementary Fig. 12b).

Due to absence of structural information of *BcATG4*, we performed *in silico* analysis to test the possibility that C158 could be a target of EB. First, we built a homology model of *BcATG4* using RoseTTAFOLD (Baek et al., 2021). Our model of *BcATG4* was superimposed with the crystal structure of *HsATG4B* (PDB, 2cy7) except two insertions parts in the N-terminal *BcATG4* (Supplementary Fig. 12a). In particular, the active site of the model of *BcATG4* is structurally conserved with that of *HsATG4B*. Next, we performed docking simulation with the model of *BcATG4* and EB using AutoDock (Morris et al., 2009). The docking simulation proposed that EB could access the active site of *BcATG4* and interact with the catalytic C158. This prediction suggests that C158 in *BcATG4* may be a target residue of EB (Supplementary Fig. 12c).

To identify additional putative target cysteines in *BcATG4*, we compared the cleavage activity of wild type and different point mutants (C to A) under mock and EB treatment. In addition to C158A, the C233A mutant among five mutants tested showed the most resistance to EB, indicating Cys233 could be a putative target residue of EB in *BcATG4* (Supplementary Fig. 12d).

In addition, since selenium-sulfur and disulfide bonds are reduced under a reducing condition (Terentis et al., 2010), we confirmed the inhibition of *BcATG4* by EB was restored under DTT treatment (Fig. 1d). We have shown the direct interaction of EB with *BcATG4* by ITC experiments (Fig. 1g). Taken together, *in silico* analysis and our experimental results suggest that EB can directly bind to putative target cysteines such as C158 and C233 in *BcATG4*. The binding of EB on the thiol group of C158 causes disturbance of the active site of *BcATG4*. Another decoration of EB on C233 appears to induce allosteric inhibition. Therefore, we postulate that the EB modification on C233 may contribute to the inhibition of the cysteine protease of *BcATG4* together with C158. The cysteine protease activity inhibited by EB causes to suppress autophagy biogenesis, resulting in attenuation of fungal virulence of *B. cinerea*.

We revised the discussion and incorporated these results (see Supplementary Fig. 12 and text lines 345-379). In the future this will open a new project to validate the precise biochemical mechanism of EB inhibition of *BcATG4*.

References:

- Baek et al., (2021). Accurate prediction of protein structures and interactions using a three-track neural network. *Science* 373, 871-876.
- Liu et al., (2019). Involvement of the cysteine protease BcAtg4 in development and virulence of *Botrytis cinerea*. *Curr Genet* 65:293-300.
- Morris et al., (2009). AutoDock4 and AutoDockTools4: Automated docking with selective receptor flexibility. *J Comput Chem* 30, 2785-2791.
- Ren et al., (2018). The autophagy gene *BcATG8* regulates the vegetative differentiation and pathogenicity of *Botrytis cinerea*. *Appl Environ Microbiol* 84:e02455-17.
- Shu et al., (2010). Synthetic substrates for measuring activity of autophagy proteases-autophagins (Atg4). *Autophagy* 6:936-947.
- Sugawara et al., (2005). Structural basis for the specificity and Catalysis of human Atg4B responsible for mammalian autophagy. *J Biol. Chem.* 280:40058-40065.
- Terentis et al. (2010). The selenazal drug ebselen potently inhibits indoleamine 2,3-dioxygenase by targeting enzyme cysteine residues. *Biochemistry* 49:591-600.

Minor Comments:

1. The manuscript mainly focused on ATG4 and ATG8. But in the third paragraph of Introduction part, there were some description of ATG1, which may interfere with readers' understanding of the manuscript. I suggest authors reconsider the importance of these description.

We revised the text according to the reviewer suggestion. We removed ATG1 parts.

2. In the Introduction, the fungicide-resistance for *B. cinerea* and *M. oryzae* is not exactly true. For *B. cinerea*, the fungicide-resistance is serious.

We thank the reviewer for pointing this out. We revised the text accordingly (lines 51-52).

3. The binding affinity between EB and ATG4 should be determined by another technology such as MST or SPR.

We are aware of the techniques that the reviewer has mentioned. Each technique has advantages and disadvantages. MST requires fluorescence labelling that may be affected by labelling efficiency and specificity. SPR needs immobilization of one of the binding partners. ITC requires large sample quantity. We reasoned that ITC is the best option for our paper because we can purify large quantity of *BcATG4* and *BcATG8* recombinant proteins. Since ITC tests if two molecules can interact or not, MST or SPR may not yield any new conclusions. Furthermore, the addition of this data will not significantly change the conclusions in the manuscript.

4. EB showed limited chemical size, did authors consider their toxicity? More importantly, I suggest authors introduce the limitations about this manuscript in the Discussion part.

For EB, clinical trials have been successfully finished and we included this in the original version of the manuscript (lines 324-328 in the revised version). The report for the clinical trial showed a dose of 400 mg of EB twice daily was effective to noise-induced hearing loss of human. This indicates that EB is edible and less toxic to humans. As per the reviewer's suggestion, we added the limitation of our HTS and the effect of EB in hosts (lines 330-343).

References:

Kil et al., (2017). Safety and efficacy of ebselen for the prevention of noise-induced hearing loss: a randomised, double-blind, placebo-controlled, phase 2 trial. *Lancet* 390:969-979.

Reviewer #2 (Remarks to the Author):

The authors use BRET-based HTS strategy to identify inhibitors of autophagy, then they validated the inhibition of EB and its analogs on spore germination, hyphal development, appressorium formation and pathogenicity of four different Ascomycota pathogens. The manuscript is well written and the authors did an impressive amount of work to validate their findings.

We appreciate reviewer's positive comments about our findings described in the manuscript.

Minor comment:

the authors have gotten the Arabidopsis transgenic plants expressing the synthetic substrate C-AtAtg8a-ShR in reference 33 and 34, why the authors did not use these plants to analysis cross reactivity of autophagy inhibitors in host cells for longer time? EB perhaps has cross inhibiting reactivity in one week or longer time.

We examined the long-term treatment of EB with the transgenic *N. benthamiana* expressing the *AtATG8*-sensor construct which was published. Our result showed the long-term treatment did not affect the processing of the *AtATG8*-sensor in the transgenic plants. These results are shown in Supplementary Fig. 11a and described in the text (lines 330-343).

Regarding the effect of EB in hosts, we discussed a possible explanation why we have not detected the inhibition of host autophagy in the discussion section (line 341-343).

Reviewer #1 (Remarks to the Author):

The manuscript presents a novel strategy for identifying chemicals that interfere with fungal autophagy, which could potentially lead to the development of new antifungal compounds. The study describes a bioluminescence resonance energy transfer (BRET)-based high-throughput screening (HTS) strategy to identify compounds that inhibit fungal ATG4 cysteine protease-mediated cleavage of ATG8, critical for autophagosome formation. The authors identified ebselen (EB) and its analogs as inhibitors of fungal pathogens' ATG4-mediated ATG8 processing, leading to a reduction in fungal pathogenicity. The research addresses a significant issue in agriculture by targeting fungal pathogens that cause substantial crop losses, emphasizing the practical implications of the study. The BRET-based HTS strategy is well-detailed and seems promising. The use of synthetic ATG8 sensors enhances specificity, and the screening method's adaptability to different species is highlighted. The findings suggest that autophagy is a viable target for developing antifungal compounds, providing potential molecular insights for the development of next-generation antifungal agents. Overall, the research is well-conducted and the manuscript is well-written. Some minor revisions could enhance clarity and readability. Here are my consolidated comments:

1. The introduction effectively sets the research context, emphasizing the economic impact of crop losses due to fungal pathogens. The transition from the problem (crop losses due to fungal pathogens) to the proposed solution (targeting autophagy in fungal pathogens) is smooth and logical. However, the structure of the sentences in the introduction should be reorganized and some word needs to be replaced. Such as lines 49 – 50 USD 10 billion 'annually'; line 54, an unmet need 'for which'; lines 58 – 59, I deleted 'it has been well known that recycling'.
2. In line 62, there seems like a sudden jump to another topic/issue. Better to find some linkers between the previous sentence and this one.
3. What other ATGs have been observed, at least mention their numbers in total so that this can be related to the other sentences (Line 64).
4. Please move "ATG genes are also conserved in the sequenced plant pathogenic fungi" in the previous paragraph. And start the sentence with "Recent studies indicate that...".
5. In line 80, write "followed by germ tube formation to find..." and provide examples of "core components of autophagy", in line 82.
6. In line 86, What are these genes (Moatg4/8)? You should provide brief information for them in the previous paragraph or sentences.
7. What do the results mentioned between lines 120 - 123 imply? Please write your interpretation.
8. The results section is concise and clearly presents the main findings of the research. However, it could benefit from simpler sentence structures. For instance, the sentence between lines 130 and 131 could be rephrased to avoid ambiguity. Please consider revising this sentence for clarity. "Our screen identified 30 and 14 compounds that inhibited and activated the BcATG4-mediated processing of BcATG8, respectively"
9. In line 143, "pharmaceutical characteristics" is a bit vague. If you're referring to the mechanism of action or pharmacodynamics, it would be clearer to state that explicitly. Also Please clarify in one sentence, why these specific analogs (EO, PT, PIO, PID) were chosen.
10. In line 158, It would be clearer to specify which functional group and which methyl modification you're referring to, this version looks a bit vague
11. Sentence between lines, 166 – 167 should be written in a simpler way, "The lowest Kd of EB with ATG4s indicates the highest affinity to both BcATG4 and MoATG4, compared to EO and PT"
12. In line 169, briefly specify, which lead compounds here you are referring to?
13. In line 213, what do you mean by Major Ascomycota pathogens, on what basis are you calling them like that? Please explain briefly in the paper.
14. In line 227, the concentrations of EB (either 5 or 10 μM) used to mix with the conidia suspension of *B. cinerea* are likely determined based on previous research or what was the basis of using this concentration?
15. In line 257, give brief information about the changes/ observations in the mutant Moatg4. maybe in 2-3 words.
16. The discussion provides a comprehensive analysis of the results and their implications. It effectively links the findings to the broader context of fungal pathogen control and autophagy research. However, please check the full text carefully and pay attention to grammatical errors and sentence structures. For example, the opening sentence in line 345 is too long, so please divide it

into two paragraphs which should be connected.

17. For the methodology section, I would recommend authors to please add references or some information behind their use of some experiment or using a certain amount of compounds.

18. In the figure legends, line 677, the author mentions "super-resolution images", in my opinion, please just write "high-resolution images" because here authors have not used TEM or other means.

19. In lines 704 – 705, please specify the aim of using grapes, strawberries, tomatoes, and rose petals as the host for fungal infections.

Point-by-point responses to reviewer comments

Reviewer #1 (Remarks to the Author):

The manuscript presents a novel strategy for identifying chemicals that interfere with fungal autophagy, which could potentially lead to the development of new antifungal compounds. The study describes a bioluminescence resonance energy transfer (BRET)-based high-throughput screening (HTS) strategy to identify compounds that inhibit fungal ATG4 cysteine protease-mediated cleavage of ATG8, critical for autophagosome formation. The authors identified ebselen (EB) and its analogs as inhibitors of fungal pathogens' ATG4-mediated ATG8 processing, leading to a reduction in fungal pathogenicity. The research addresses a significant issue in agriculture by targeting fungal pathogens that cause substantial crop losses, emphasizing the practical implications of the study. The BRET-based HTS strategy is well-detailed and seems promising. The use of synthetic ATG8 sensors enhances specificity, and the screening method's adaptability to different species is highlighted. The findings suggest that autophagy is a viable target for developing antifungal compounds, providing potential molecular insights for the development of next-generation antifungal agents. Overall, the research is well-conducted and the manuscript is well-written. Some minor revisions could enhance clarity and readability.

We thank the reviewer for a positive note on our paper and providing suggestions to improve the readability of the paper.

Here are my consolidated comments:

1. The introduction effectively sets the research context, emphasizing the economic impact of crop losses due to fungal pathogens. The transition from the problem (crop losses due to fungal pathogens) to the proposed solution (targeting autophagy in fungal pathogens) is smooth and logical. However, the structure of the sentences in the introduction should be reorganized and some word needs to be replaced. Such as lines 49 – 50 USD 10 billion ‘annually’; line 54, an unmet need ‘for which’; lines 58 – 59, I deleted ‘it has been well known that recycling’.

Revised as suggested by the reviewer; see lines 53, 57, and 64.

2. In line 62, there seems like a sudden jump to another topic/issue. Better to find some linkers between the previous sentence and this one.

As suggested by the reviewer, we added a sentence for better connection. See lines 60-61.

3. What other ATGs have been observed, at least mention their numbers in total so that this can be related to the other sentences (Line 64).

We added a sentence to reflect the reviewer suggestion, see lines 68-69.

4. Please move “ATG genes are also conserved in the sequenced plant pathogenic fungi” in the previous paragraph. And start the sentence with “Recent studies indicate that...”.

We have revised accordingly. See line 84.

5. In line 80, write “followed by germ tube formation to find...” and provide examples of “core components of autophagy”, in line 82.

Revised according to the reviewer suggestion; see line 87. In general, autophagy-related genes characterized in yeast are considered core components. Higher eukaryotes have other proteins that play a role in autophagy in addition to the core components. Please see answer to #3 comment.

6. In line 86, What are these genes (*Moatg4/8*)? You should provide brief information for them in the previous paragraph or sentences.

Moatg4 and *Moatg8* are mutants of *ATG4* and *ATG8* genes in *M. oryzae*. We followed the nomenclature of mutants from the cited references. See lines 92-93.

7. What do the results mentioned between lines 120 - 123 imply? Please write your interpretation.

We added the interpretation for the result. Please see lines 133-136.

8. The results section is concise and clearly presents the main findings of the research. However, it could benefit from simpler sentence structures. For instance, the sentence between lines 130 and 131 could be rephrased to avoid ambiguity. Please consider revising this sentence for clarity. “Our screen identified 30 and 14 compounds that inhibited and activated the BcATG4-mediated processing of BcATG8, respectively”

We revised the sentence. See lines 148-149.

9. In line 143, “pharmaceutical characteristics” is a bit vague. If you’re referring to the mechanism of action or pharmacodynamics, it would be clearer to state that explicitly. Also Please clarify in one sentence, why these specific analogs (EO, PT, PIO, PID) were chosen.

We removed “pharmaceutical characteristics” and revised the sentence. See lines 155-158.

10. In line 158, It would be clearer to specify which functional group and which methyl modification you’re referring to, this version looks a bit vague

We specified the functional group. See lines 169-171.

11. Sentence between lines, 166 – 167 should be written in a simpler way, “The lowest Kd of EB with ATG4s indicates the highest affinity to both BcATG4 and MoATG4, compared to EO and PT”

We revised the sentence. See lines 179-180.

12. In line 169, briefly specify, which lead compounds here you are referring to?

We changed “lead compounds” to “EB and its analogs”; see line 181.

13. In line 213, what do you mean by Major Ascomycota pathogens, on what basis are you calling them like that? Please explain briefly in the paper.

In lines 123-124, we stated that “*B. cinerea* and *M. oryzae* are among the top ten devastating fungal pathogens³⁶”. However, *Sclerotinia sclerotiorum* and *Monilinia fructicola* are not included in the top ten list. Therefore, in the revised version we removed the word “major”. See line 229 and lines 231-234.

14. In line 227, the concentrations of EB (either 5 or 10 μM) used to mix with the conidia suspension of *B. cinerea* are likely determined based on previous research or what was the basis of using this concentration?

The reason we chose the concentrations is based on our experimental result of the germination test of *B. cinerea* shown in Fig. 3b. We modified the sentence to reflect this. See lines 244-247.

15. In line 257, give brief information about the changes/ observations in the mutant Moatg4. maybe in 2-3 words.

We revised according to the reviewer suggestion. See lines 269-271.

16. The discussion provides a comprehensive analysis of the results and their implications. It effectively links the findings to the broader context of fungal pathogen control and autophagy research. However, please check the full text carefully and pay attention to grammatical errors and sentence structures. For example, the opening sentence in line 345 is too long, so please divide it into two paragraphs which should be connected.

We have carefully checked the manuscript and fixed the sentence structures and grammatical errors. As suggested by the reviewer, we have split the first paragraph of the discussion section into two paragraphs. The paragraph is split at line 337 as suggested by the reviewer.

17. For the methodology section, I would recommend authors to please add references or some information behind their use of some experiment or using a certain amount of compounds.

We have carefully revised the Method section. We added specific amount of compounds used as applicable.

18. In the figure legends, line 677, the author mentions “super-resolution images”, in my opinion, please just write “high-resolution images” because here authors have not used TEM or other means.

The name of this microscopic technology (AiryScan) from the developer is “super resolution confocal microscopy”. Thus we will keep the name as is; see lines 899-902.

19. In lines 704 – 705, please specify the aim of using grapes, strawberries, tomatoes, and rose petals as the host for fungal infections.

We revised the text; see lines 296-298.